# Tinnitus risk factors and its evolution over time

Lise Hobeika [1,2,3,4] ✉, Matt Fillingim[3,5], Christophe Tanguay-Sabourin [3,6,7], Mathieu Roy[3,4,6], Alain Londero [8], Séverine Samson[1,9,10] & Etienne Vachon-Presseau [3,6,11]

Subjective tinnitus is an auditory percept unrelated to external sounds, for which the limited understanding of its risk factors complicates the prevention and management. In this study, we train two distinct machine learning models to predict tinnitus presence (how often individuals perceive tinnitus) and severity separately using socio-demographic, psychological, and health-related predictors with the UK Biobank dataset (192,993 participants, 41,042 with tinnitus). We show that hearing health was the primary risk factor of both presence and severity, while mood, neuroticism, and sleep predicted severity. The severity model accurately predicts tinnitus progression over nine years, with a large effect size for individuals developing severe tinnitus (Cohen's d = 1.3, ROC = 0.78). This result is validated on 463 individuals from the Tinnitus Research Initiative database. We simplify the severity model to a six-item clinical questionnaire that detects individuals at risk of severe tinnitus, for which early supportive care would be crucial.

Subjective tinnitus is an auditory symptom characterized by the perception of sound without any external acoustic stimulus[1]. This symptom is common, with a prevalence of ~14% in the general population, but its severity is highly variable[2–4]. Tinnitus is not bothersome for most individuals, but it is highly distressing for others who experience associated sleep disorders, socio-emotional disturbances (i.e., anxiety, depression), and cognitive difficulties[5]. As there is no cure to eliminate tinnitus perception—only palliative interventions aiming at reducing associated distress[1,6,7]-improving tinnitus prevention and clinical management by identifying the key associated risk factors is crucial.

Tinnitus is thought to arise from a maladaptive reaction to auditory peripheral damage caused by factors such as presbycusis, noise exposure, ototoxic medication, or trauma, leading to sensory deafferentation[8]. This deafferentation disrupts auditory inputs along the auditory pathway, prompting the nervous system to generate the perception of sound in the absence of an external stimulus[9]. In this context, tinnitus is considered a phantom sound, analogous to phantom limb pain, where individuals perceive sensations in a missing limb due to maladaptive neural activity in the brain. However, this pathophysiological explanation remains unsatisfactory as not everyone with hearing loss experiences tinnitus, nor does it adequately account for the distress associated with tinnitus. Observed discrepancies in tinnitus experiences may be due to variations in sociodemographic, psychological, hearing, or physical health[10–12]. Moreover, emotional and sleep disorders, often seen as consequences of tinnitus, may also be risk factors for its appearance or severity. In this case, psychosocial factors may instead contribute to shaping how tinnitus is experienced by the patient.

[1]Université Paris Cité, Institut Pasteur, AP-HP, INSERM, CNRS, Fondation Pour l'Audition, Institut de l'Audition, IHU reConnect, Paris, France. [2]Sorbonne Université, Institut du Cerveau—Paris Brain Institute—ICM, INSERM, CNRS, APHP, Hôpital de la Pitié Salpêtrière, Paris, France. [3]Alan Edwards Centre for Research on Pain, McGill University, Montreal, QC, Canada. [4]Department of Psychology, McGill University, Montreal, QC, Canada. [5]Integrated Program in Neuroscience, McGill University, Montreal, QC, Canada. [6]Department of Anesthesia, Faculty of Medicine and Health Sciences, McGill University, Montreal, QC, Canada. [7]Faculty of Medicine, Université de Montréal, Montreal, QC, Canada. [8]Université Paris Cité, Institut Pasteur, AP-HP, Hôpital Lariboisière, Service ORL, INSERM, Fondation Pour l'Audition, IHU reConnect, Paris, France. [9]PSITEC – Psychologie: Interactions, Temps, Emotions, Cognition, Université de Lille, Lille, France. [10]Epilepsy Unit, Assistance Publique – Hôpitaux de Paris, Pitié-Salpêtrière Hospital, Paris, France. [11]Faculty of Dental Medicine and Oral Health Sciences, McGill University, Montreal, QC, Canada. ✉e-mail: lise.hobeika@gmail.com

A longitudinal examination of the risk factors predicting the different facets of tinnitus is currently lacking. The objective of this study is to identify the risk factors predicting the onset and evolution of tinnitus over time. To this aim, we applied machine learning algorithms to data from the UK Biobank dataset. This extensive biomedical database contains detailed longitudinal information on lifestyle, socio-economic background, hearing, physical, and mental health from over 190,000 individuals. As tinnitus presence is not necessarily associated with severity[13], we analyzed data with two distinct models to predict i) tinnitus presence and ii) tinnitus severity, using a pipeline recently developed to study chronic pain[14]. We also used the Tinnitus Research Initiative (TRI) dataset to validate the models trained on the UK Biobank[14].

## Results

### Descriptive

Table 1 shows the prevalence of tinnitus presence and the severity categorization among participants with tinnitus. The percentage of participants with tinnitus was 21.3%, with 22.7% of them experiencing moderate or severe distress. Tinnitus presence was more prevalent in men, and its prevalence increased with age, whereas women reported experiencing greater distress from tinnitus. Tinnitus presence and severity were associated with larger hearing deficits, be it self-reported hearing difficulties, speech-in-noise hearing difficulties self-reported and measured, or use of hearing aid or cochlear implants (see Table 1 for statistics).

### Tinnitus presence risk score: classification at baseline and evolution in time

**Risk score calculation.** We first used the nonlinear iterative partial least squares (NIPALS) algorithm to train a model predicting four different levels of tinnitus severity: no tinnitus, some of the time, a lot of the time, or most or all the time. The weights of each feature in the model are presented in Fig. 1A and their loadings are presented in Supplementary Fig. S1. The following results show the model performance tested in participants from the held-out test set. The model explained a total of 12.5% of the variance of tinnitus presence, with the most explained variance coming from hearing health (12.5%) followed by demographic (2.1%), whereas other categories explained the least variance (<1% each; Fig. 1B). The risk score for tinnitus presence showed good to excellent performance for classifying participants with tinnitus from tinnitus-free participants, as shown by their diagnostic capacities (AUC 0.68–0.80, Fig. 1C) and effect sizes (Cohen's $d = 0.67$–1.28). Retraining the model after removing each category of features showed that features from hearing health were the most important for the prediction of tinnitus presence (Fig. 1D). We confirmed the validity of the risk score independently for women and men (ROC-AUC > 0.78 for all the time vs no tinnitus, for both men and women; Supplementary Fig. S2), and for different ethnicities despite smaller sample sizes (ROC-AUC > 0.80 for all the time vs no tinnitus, for Asian, Black, and White ethnicities; Supplementary Fig. S3A).

**Recovery and worsening over time: 9-year prognosis.** Participants evolution of tinnitus presence at the two visits are displayed in Fig. 1E (supplementary Table S1 for the detailed numbers, and Supplementary Fig. S4 for Odds ratios representing those evolutions). Risk scores were adjusted to account for baseline presence to focus on the changes, recovery, or worsening, over time (Supplementary Fig. S5). The adjusted presence risk score did not predict the evolution of tinnitus at the follow-up visit, as evidenced by the Cohen's $d$ (all < 0.40) and AUC-ROC levels (AUC < 0.58, Fig. 1F, G). As the presence risk score did not predict the evolution, we trained a new model specifically to predict the evolution of tinnitus presence over time, using the 101 features. This new model was also unsuccessful at predicting the evolution of tinnitus presence over time (AUC-ROC ≤ 0.60 for every evolution level). This suggests that the evolution of tinnitus presence is difficult

**Table 1 | Baseline characteristics of participants of the UK Biobank**

| | Tinnitus presence (N = 192,993) | | | | | Tinnitus severity (N = 41,042) | | | | |
| --- | --- | --- | --- | --- | --- | --- | --- | --- | --- | --- |
| | Never had tinnitus | Tinitus some of the time | Tinitus a lot of the time | Tinitus all the time | p value | No distress | Mild distress | Moderate distress | Severe distress | p value |
| Prevalence | 78.7% | 10.4% | 3.0% | 7.9% | | 29.0% | 48.3% | 19.0% | 3.7% | |
| Demographics | | | | | | | | | | |
| Sex (% M/F) | 44.2/55.8 | 48.6/51.4 | 53.4/46.6 | 59.5/40.5 | <10^-300 | 55.9/44.1 | 48.8/51.2 | 46.9/53.1 | 44.4/55.6 | 10^-78 |
| Age | 58.1 (8.7) | 59.7 (8.3) | 60.8 (7.8) | 61.4 (7.3) | <10^-300 | 58.9 (6.6) | 59.2 (6.2) | 59.1 (6.5) | 58.5 (6.8) | n.s. |
| Hearing health | | | | | | | | | | |
| Self-reported hearing difficulties | 20.1% | 42.6% | 55.8% | 67.9% | <10^-300 | 43.9% | 53.2% | 67.4% | 76.7% | <10^-300 |
| Self-reported speech-in-noise hearing difficulties | 31.0% | 52.3% | 61.2% | 67.2% | <10^-300 | 51.8% | 61.1% | 71.9% | 78.5% | 10^-223 |
| Speech reception threshold in dB | −6.4 (1.8) | −6.1 (2.1) | −5.9 (2.1) | −5.9 (2.2) | <10^-300 | −6.2 (1.9) | −6.1 (1.9) | −5.9 (2.2) | −5.5 (2.7) | 10^-50 |
| Hearing aid users | 2.6% | 6.6% | 8.5% | 14.1% | <10^-300 | 8.0% | 8.6% | 13.2% | 19.8% | 10^-59 |
| Cochlear implants | 0.054% | 0.25% | 0.17% | 0.22% | 10^-18 | 0.18% | 0.19% | 0.32% | 0.72% | 10^-3 |

Data are percentages, except for the age and speech reception Threshold for which the mean and standard deviation are given. Group differences were tested with a Chi² test for the sex proportions, and with an ANOVA for the Age and hearing health features.

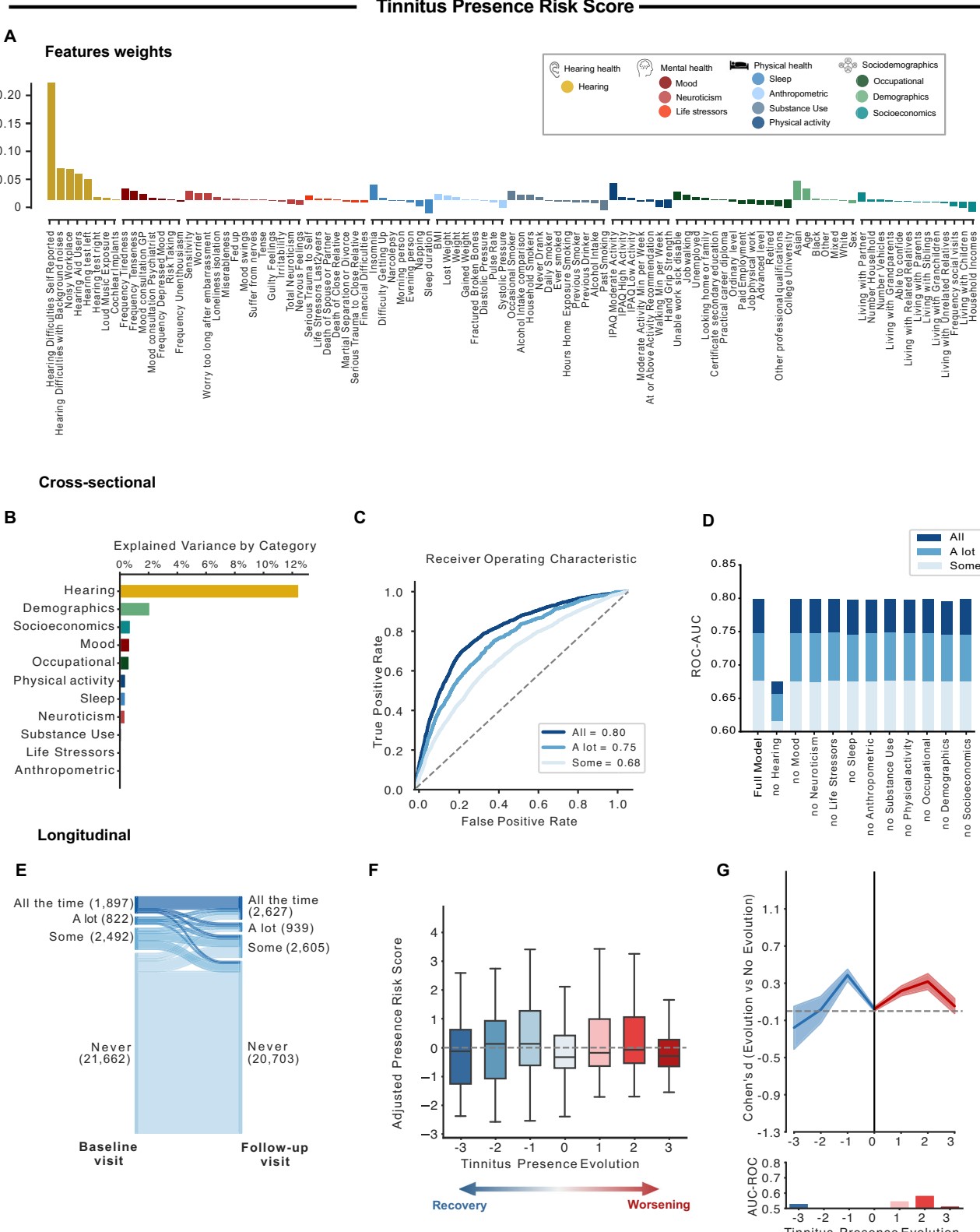

**Tinnitus Presence Risk Score**

**A** Features weights

Cross-sectional

**B** Explained Variance by Category

**C** Receiver Operating Characteristic

**D**

Longitudinal

**E** Baseline visit — Follow-up visit

**F** Adjusted Presence Risk Score / Tinnitus Presence Evolution — Recovery / Worsening

**G** Cohen's d (Evolution vs No Evolution) / AUC-ROC / Tinnitus Presence Evolution

to predict based on general health, sociodemographic, or environmental factors.

**Tinnitus severity risk score: classification at baseline and evolution in time**

**Risk score calculation.** We next used NIPALS algorithm to train a model predicting four different levels of tinnitus severity: not at all,

mild, moderate, or severe. The weights of each feature in the model are presented in Fig. 2A and their loadings are presented in Supplementary Fig. S1. The model explained a total of 9.2% of the variance of the tinnitus severity, with the most explained variance coming from neuroticism (3.7%), hearing health (3.6%), mood (3.0%), and sleep (2.2%), whereas other categories explained the least variance (<1% each; see Fig. 2B). The risk score for tinnitus severity showed moderate to

**Fig. 1 | Tinnitus presence model. A** We used the NIPALS machine learning algorithm to predict the presence of tinnitus based on 101 features, representing eleven categories. The weights attributed to the features are depicted in **A**. The model was trained on 166,119 participants and tested on 26,874 individuals. **B**, **C** tested the efficacy of the model categorizations at baseline in the test set, while **F**, **G** evaluated the model's predictions of tinnitus evolution over time (9 years after baseline). **B** This figure depicts the variance explained by each category of the model. Only hearing and demographic factors explained >1% of the variance each. **C** We used the AUC-ROC curves to test if the model was able to categorize participants based on how often they experienced tinnitus. The model predictions were good to excellent, depending on the level (some of the time, a lot of the time, all the time). **D** The model was re-trained removing each category of features. Only the removal of hearing health significantly impacted the performance. **E** This panel depicts the evolution of tinnitus presence between the baseline visit (left side) and the follow-up visit (right side) of participants of the testing dataset, spaced by nine years. **F**, **G** These panels illustrated the evolution of the adjusted risk scores (the boxplot depicts median (center line), interquartile range (box), and 1.5 × IQR whiskers) (**F**) and the model's performances (effect sizes calculated using Cohen's $d$ and categorization efficacy assessed with AUC-ROC (**G**)) as a function of tinnitus presence over time. The 95% CI estimated across 10,000 bootstrap samples is shown for the effect size. The evolution is rated between −3 and 3, with −3 representing the evolution from tinnitus present all the time at baseline to the absence of tinnitus in the follow-up visit (full recovery), and +3 the opposite evolution (apparition of constant tinnitus). Based on those figures, we concluded that the model could not predict the evolution of tinnitus presence over time.

excellent performance for classifying participants with distressing tinnitus from distress-free participants, as shown by diagnostic capacities (AUC 0.61–0.78, Fig. 2B) and their effect sizes (Cohen's $d$ = 0.39–1.15). Retraining the model after removing each category of features showed the multifactorial nature of tinnitus severity, as not a single category was essential to the model performance. Thus, the model accuracy decreased only after removing the four identified categories (hearing, mood, neuroticism and sleep), evidencing the important of a holistic model for the prediction of tinnitus severity (Fig. 2D). We confirmed the validity of the risk score independently for women and men (ROC-AUC > 0.77 for distressing tinnitus from distress-free participants, for men and women; Supplementary Fig. S2), and different ethnicities (ROC-AUC > 0.77 for severe tinnitus vs no distress, for Asian, Black and White ethnicities, Supplementary Fig. S3B).

**Recovery and worsening over time: 9-year prognosis.** The stability and individual changes in tinnitus severity between the two visits are displayed in Fig. 2E (supplementary Table S2 for the detailed numbers, and Fig. S4 for Odd ratios representing those evolutions). Here again, risk scores were adjusted to account for baseline severity to predict recovery or worsening over time (See Supplementary Fig. S5). The adjusted severity risk score predicted the evolution of tinnitus at the follow-up visit, as seen in the evolution plot (Fig. 2F), the Cohen's d and AUC-ROC levels (AUC = 0.62 for an evolution from severely distressing to no distress, and AUC = 0.81 for an evolution from no distress to severely distressing, Fig. 2G).

**Evolution of tinnitus severity over time: a clinical questionnaire.** Last, we aimed to simplify our model and reduce the number of features by extracting those with the highest predictive value. This simplified model represents a reduced risk score for tinnitus severity calculated by simply summing the binarized answers to six items measuring hearing health, sleep, neuroticism, and mood, selected with a linear forward feature selection algorithm. The resulting questionnaire, comprising these six items is called POST (Prediction Of the Severity of Tinnitus; Fig. 3A). Based on the odd ratios of experiencing no, mild, moderate or severe at the follow-up visit depending on the Simplified risk score at baseline, we concluded that scores of 0 and 1 are associated with a low risk, 2 and 3 are associated with a moderate risk, 4 and 5 are associated with high risk and 6 with a very high risk of experiencing moderate of severe distress over time (see Fig. 3B and Supplementary Fig. S6). The adjusted simplified risk score had an average to good performance in predicting tinnitus evolution at the baseline visit (Fig. 3C) and at the longitudinal dataset (Fig. 3D, E), especially in the prediction of individuals who will evolve from a non-distressing tinnitus to a severely distressing one (Cohen's $d$ = 1.3, ROC = 0.78). This represented a good trade-off between the sparsest number of features and the highest AUC-ROC.

We validated our simplified risk score on 463 patients experiencing tinnitus in the TRI dataset. Even though the Severity was evaluated

with 5 categories, the evolution was rated between −3 and 3, and not −4 and 4, as there were no individuals who evolved from no distress to an invalidating tinnitus or vice versa. Here, the simplified risk score associated with tinnitus severity achieved good performance (Fig. 3F), concordant with what was initially observed in the UK Biobank. POST also had an average to good performance to predict tinnitus evolution in the longitudinal dataset (Fig. 3G, H), especially in the prediction of individuals who will evolve from a non-distressing tinnitus to a severely distressing one (Cohen's $d$ = 2.5, ROC = 0.94).

## Discussion

The objective of this study was to identify the risk factors of tinnitus presence and severity, as well as their evolution over time. The results revealed a dissociation between the features predicting tinnitus presence and tinnitus severity. While hearing health emerged as a common key predictor of presence and severity, mood, neuroticism, and sleep only predicted its severity. Interestingly, while the presence model did not predict the evolution of tinnitus over time, the severity model provided an estimation of its progression over nine years, with a large effect size for individuals who develop severe tinnitus. A simplified version of the risk score for tinnitus severity was derived from five questions with binarized outcomes and validated in an independent cohort with the aim of detecting individuals at risk of developing severe tinnitus over time.

In the UK Biobank, tinnitus prevalence was 21.3%, with a moderate to severe distress for 21.9% of them, which is in line with the common prevalence observed for this age range (40–70)[2,15]. Tinnitus was more prevalent in men than in women, while tinnitus severity was higher in women. Looking at the relationship between tinnitus and all measures of hearing health, we observed increasing deficits with increasing tinnitus presence and with increasing tinnitus severity. Overall, our results indicated that, even if the UK Biobank has potential biases such as healthy volunteer selection bias[16], recall bias, lack of test-retest on a single time point in the newly developed questionnaire, the prevalence of tinnitus and the odds ratio for its evolution are aligned with the literature for this age range[2,9,17–19].

Various factors have been associated with tinnitus presence or severity, such as mental health, education level, chronotype, physical exercise or alcohol consumption[20–22], but with low levels of evidence[11]. These often interrelated factors are usually studied in isolation. To overcome this limitation, we included a large variety of possible risk factors in the same multivariate model, covering sociodemographics, hearing health, mental health, and physical health factors, merging them into categories to create a global picture of tinnitus pathophysiology. First, we showed that the major predictor of tinnitus presence is hearing health, and in particular, self-reported hearing difficulties. It confirms the large literature pointing toward hearing deficits as the main risk factor for tinnitus apparition[1,23]. The second identified risk factor is age, which is likely mediated by presbycusis. Emotional factors have also been identified as potential risk factors in the literature[2], attributed to so-called

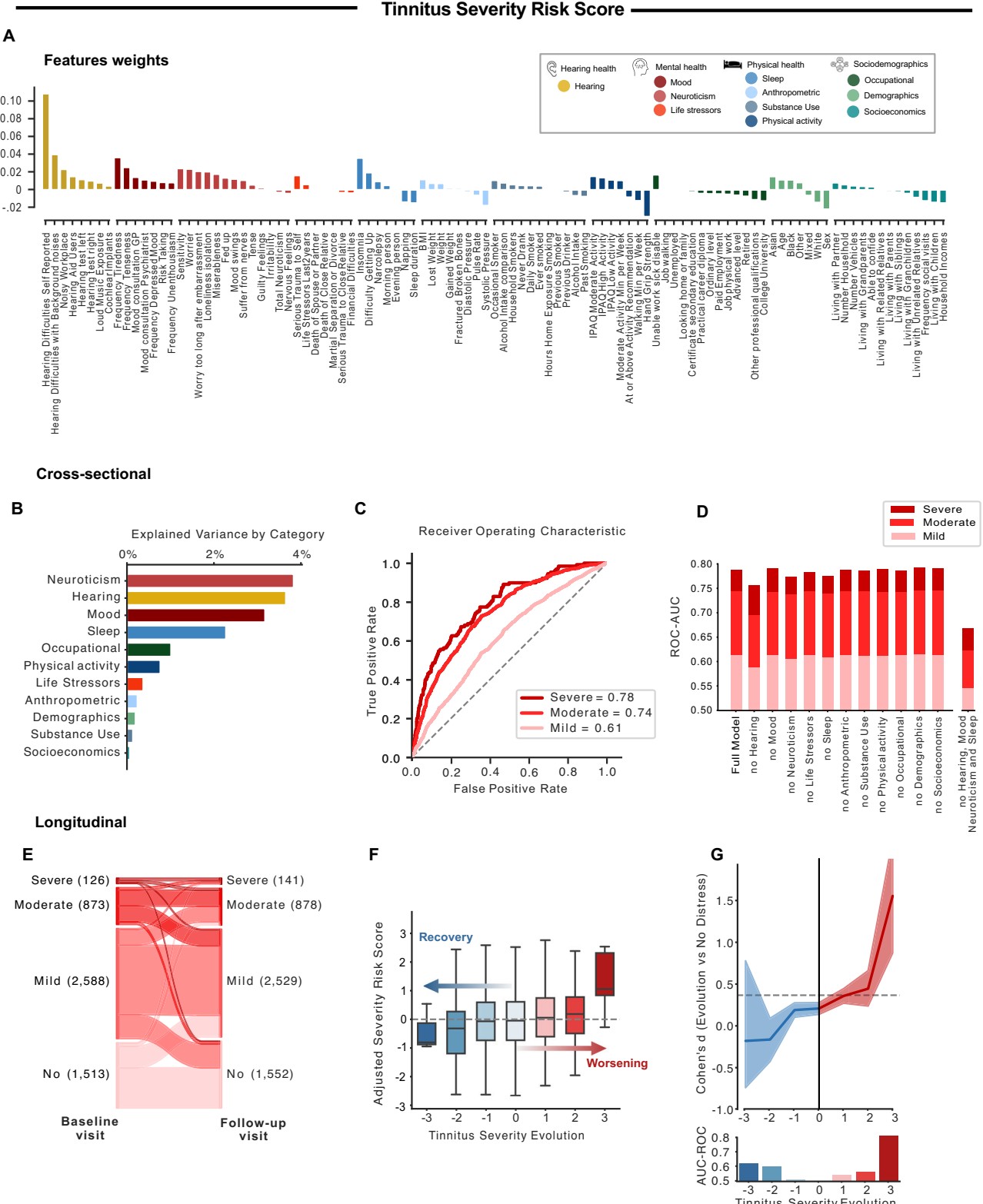

stress-induced tinnitus[24]. Our results suggest that emotional factors explain only a small part of the variance, showing limited predictive capacities. Physical health factors, like anthropometrics, physical activity, and substance use, had no predictive value for tinnitus presence. Overall, our results indicate that tinnitus presence is mainly predicted by hearing health. They do not explain the fact that not all individuals with hearing loss develop tinnitus. In order to clarify this observation, it is essential to look into biological factors

such as genetics[25] and cerebral functioning[26], which extend beyond the scope of this study.

Our results show that tinnitus severity is predicted mainly by: mood (anxiety, depression), neuroticism (i.e., personality trait characterized by a tendency to respond with negative emotions to threat, frustration, or loss[27]), sleep, and life stressors. This result is in line with the literature that has extensively associated severe tinnitus with stress, depression, personality traits, and sleep disorders[1,5,12,28]. On the

**Fig. 2 | Tinnitus severity model. A** We used the NIPALS machine learning algorithm to predict the severity of tinnitus based on 101 features, representing eleven categories. The weights attributed to the features are depicted in **A**. The model was trained on 35,942 participants and tested on 5100 individuals. **B, C** tested the efficacy of the model categorizations at baseline in the test set, while **F, G** evaluated the model's predictions of tinnitus evolution over time (9 years after baseline). **B** This figure depicts the variance explained by each subcategory of the model separately. The mood, neuroticism, hearing, and sleep explained >1% of the variance each (**C**). We used the AUC-ROC curves to test if the model was able to categorize participants based on tinnitus severity. The model predictions were good to excellent, depending on the level (mild, moderate, or severe distress). **D** The model was re-trained removing each category of features, and the four identified categories (hearing, mood, neuroticism, and sleep). Only the removal of the four categories significantly impacted the performances. **E** This panel depicts the evolution of tinnitus severity between the baseline visit (left side) and the follow-up visit (right side) of participants of the testing dataset, spaced by nine years. **F, G** These panels illustrated the evolution of the adjusted risk scores (the boxplot depicts the median (center line), interquartile range (box), and $1.5 \times IQR$ whiskers) (**F**) and the model's performances (effect sizes calculated using Cohen's $d$ and categorization efficacy assessed with AUC-ROC (**G**)) as a function of tinnitus severity over time. The 95% CI estimated across 10,000 bootstrap samples is shown for the effect size. The evolution is rated between −3 and 3, with −3 representing the evolution from severe distress to no distress, and +3 the opposite evolution. Based on those figures, we concluded that the model had good performance in predicting the evolution of tinnitus over time for the −3 and +3 categories, with excellent performance in identifying participants who will develop severe distress.

other hand, the association between severity and hearing health has been reported more rarely in the literature[29,30]. This could reflect differences in tinnitus loudness or masking level. Other potential risk factors identified in the literature, like physical activity or substance use, show very small predictive value. We hypothesized that these effects are mediated by other socio-emotional factors.

After identifying risk factors, our main challenge was to understand the factors mediating tinnitus evolution over time. The risk score for tinnitus presence was unable to predict the evolution in the levels of tinnitus presence over time, which was expected, as the risk score primarily reflects hearing deficits that may either not have yet occurred or are largely irreversible. This suggests that risk factors for the evolution of tinnitus presence may be the result of pathophysiological mechanisms in the auditory periphery or the central nervous system rather than from psychosocial factors. On the contrary, we showed that mood, personality traits, sleep, and hearing dysfunction were the strongest predictors of the evolution of tinnitus severity over time. We evidenced this effect after controlling for participants' distress risk scores at baseline, demonstrating that the predictions are not merely based on the relationship between baseline state and follow-up state. Instead, we investigated how differences between participants' severity levels and their baseline risk scores could serve as a prognosis of their evolution over time. These results align with previous studies associating tinnitus severity with mood and sleep disorders, but for the first time, we demonstrate that these factors actually predispose tinnitus severity evolution, evidencing a key role of those factors in the prediction of its progression. This implies that these factors, by means of sound or psychologically oriented therapies, may not stop tinnitus perception per se but may instead alleviate how they are experienced. Those results are in coherence with the clinical approach of decreasing the severity instead of stopping the perception.

To improve clinical utility, we developed a 6-item questionnaire to predict tinnitus severity over time. Our findings show that higher scores on the questionnaire were associated with a larger odds ratio of developing severe tinnitus in the future. These results were validated using an independent clinical dataset (TRI database, 462 participants), ensuring the generalizability of the findings. Overall, this tool represents an easy-to-use prognostic resource for identifying patients unlikely to habituate to tinnitus. This questionnaire has the potential to be a key tool in improving tinnitus clinical management. Clinical resources for tinnitus management are limited, including the number of ENTs specialized in tinnitus and therapists trained in cognitive behavioral therapy[31]. Clinicians in primary care could use our questionnaire to help in their decisions to refer patients to tinnitus specialists. Additionally, specialists could use the tool to focus their clinical and therapeutic efforts on patients at higher risk of developing severe tinnitus, avoiding unnecessary or excessive interventions for those likely to habituate. This concern is particularly relevant in common clinical scenarios, such as idiopathic sensorineural hearing loss, the complex aftermath of otological surgery, or newly onset tinnitus.

Further clinical studies are necessary to validate the efficacy of this tool across various clinical populations.

Our study has several limitations. First, the UK Biobank lacks ethnic diversity, with 91% of our sample being of White descent. This limitation may introduce bias into our models, potentially leading to a mischaracterization of non-White participants[32]. Even if we confirmed that our models performed well for individuals of Asian and Black ethnicities (Supplementary Fig. S6), replication in datasets with greater ethnic diversity is necessary to validate our findings. Second, the assessment of tinnitus presence lacks granularity in the UK Biobank, as it does not include a pure constant tinnitus category. Additionally, the absence of hyperacusis evaluation—a hypersensitivity to noise commonly co-occurring with tinnitus—limits our ability to account for this important comorbidity[33]. Third, adding important biological factors in our analysis, such as genetic and cerebral contributions[34,35], should be considered in the future, especially for predicting the presence of tinnitus, as recent studies highlight the significant role of genetics. For instance, Clifford et al. identified specific genetic architectures that differentiate the perception of tinnitus from hearing loss[34]. Additionally, evidence from neuroimaging studies points to distinct patterns of brain functioning associated with tinnitus[36,37]. Moreover, Edvall et al.[9] recently demonstrated that altered auditory brainstem responses distinguish constant tinnitus from occasional tinnitus. These findings underscore the importance of incorporating a broader range of environmental and biological factors in future studies. Finally, we developed a tool to identify individuals at risk of developing severe tinnitus; however, further research is needed to determine the most effective treatments to prevent this debilitating condition in those patients.

Overall, our study clearly distinguishes between the presence and severity of tinnitus. Tinnitus presence, associated with hearing health, highlights the necessity for raising public awareness about the irreversible consequences of peripheral auditory damage induced by noise or ototoxic drugs. Conversely, tinnitus severity is influenced by psychosocial factors, underscoring the significance of interventions targeting these factors. It is also influenced by hearing health, suggesting that deeper hearing loss will trigger more intrusive tinnitus. Additionally, we show that only tinnitus severity can be predicted, pinpointing differences in risk factors associated with each dimension (presence and severity).

## Methods
### UK Biobank dataset
The UKB dataset is a comprehensive and forward-looking collection of data. More details can be found at https://www.ukbiobank.ac.uk/media/gnkeyh2q/study-rationale.pdf.

**Participants.** Participants aged between 40 and 69 years old, who consented to participate in the study, underwent a first evaluation at one of the 22 assessment centers in UK. A subset of the participants

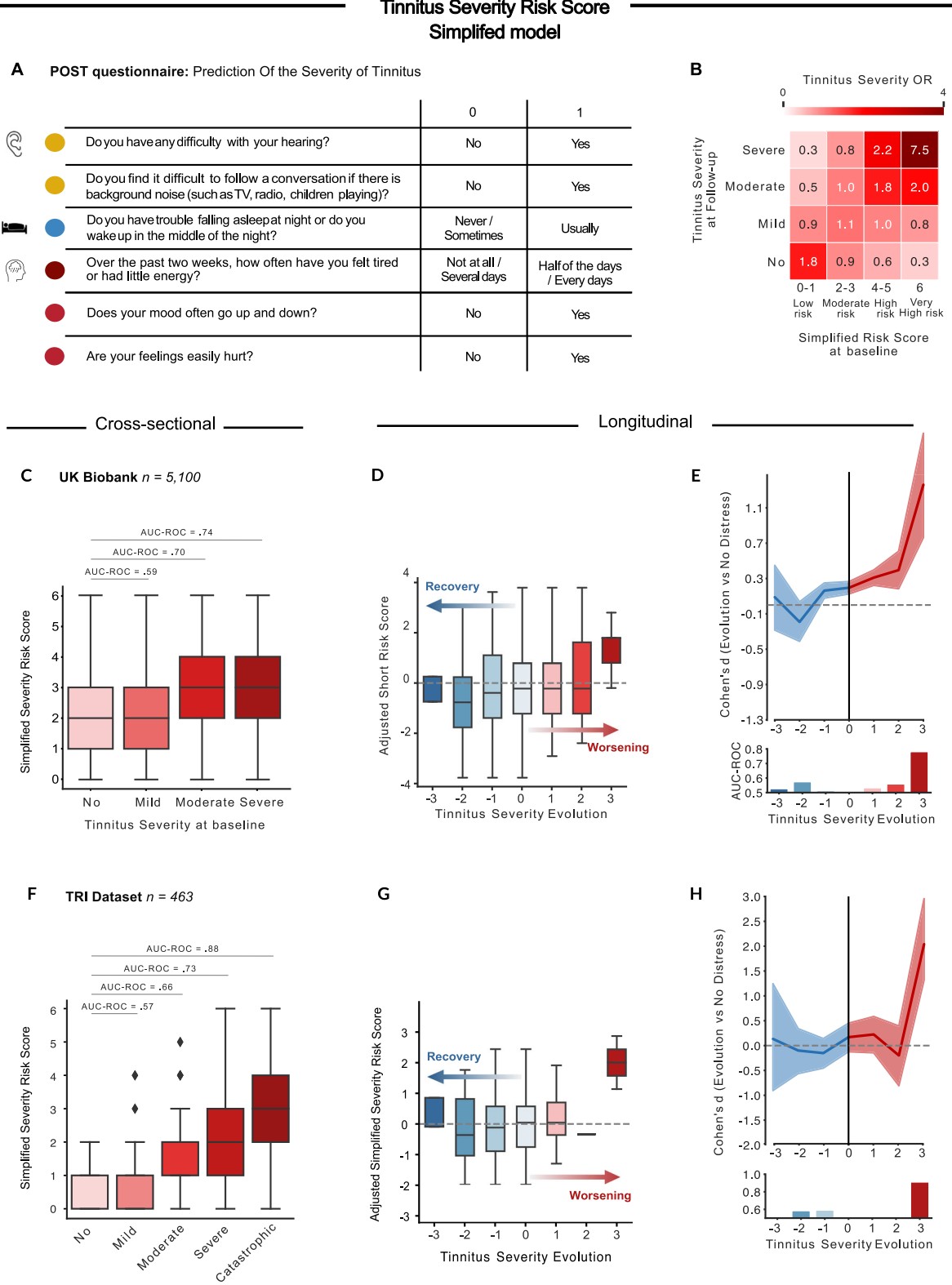

**Figure (A–H): Tinnitus Severity Risk Score — Simplified model**

**A** — POST questionnaire: Prediction Of the Severity of Tinnitus

| | 0 | 1 |
|---|---|---|
| Do you have any difficulty with your hearing? | No | Yes |
| Do you find it difficult to follow a conversation if there is background noise (such as TV, radio, children playing)? | No | Yes |
| Do you have trouble falling asleep at night or do you wake up in the middle of the night? | Never / Sometimes | Usually |
| Over the past two weeks, how often have you felt tired or had little energy? | Not at all / Several days | Half of the days / Every days |
| Does your mood often go up and down? | No | Yes |
| Are your feelings easily hurt? | No | Yes |

**B** — Tinnitus Severity OR

Cross-sectional (C, F) and Longitudinal (D, E, G, H) panels.

**C** UK Biobank n = 5,100
**F** TRI Dataset n = 463

was subsequently invited for follow-up visits. We used data from all the visits to constitute the baseline (V1, collected between April 2009 and November 2021) and the follow-up (V2, collected between August 2012 and February 2023) datasets. More information about the constitution of those datasets is available in the Supplementary Fig. S7 and Table S3.

**Tinnitus phenotypes in the UK Biobank.** Tinnitus presence was assessed by: "Do you get or have you had noises (such as ringing or buzzing) in your head or in one or both ears that last for more than five minutes at a time?". Answers were: *(1) Yes. now most or all of the time, (2) Yes. Now, a lot of the time, (3) Yes. Now, some of the time, (4) Yes, but not now. But have in the past, (5) No. never, (6) Do not know, (7) Prefer not to*

**Fig. 3 | Validation of the POST questionnaire. A** Based on the severity model, we extracted six items explaining the most variance to create a short questionnaire, with binarized answers to estimate the risk of developing moderate or severe tinnitus over time. The simplified risk score consists of two items on hearing health, one on sleep disorders, and three on mental health. **B** This figure depicts the odd ratios (OR) of experiencing no, mild, moderate or severe distress associated with tinnitus at a follow-up visit based on the risk score at Baseline. Those odd ratios were calculated on the testing dataset of the UK Biobank. Based on them, we observed that a risk scores of 0 and 1 is associated with a low risk of developing moderate or severe distress over time, risk scores of 2 and 3 are associated with a moderate risk, risk scores of 4 and 5 are associated with a high risk and risk scores of 6 are associated with a very high risk. **C–G** We tested the validity of this simplified risk score on data on 5317 participants of the UK Biobank (**B**–**D**) and on 467 participants of the TRI (**E**–**G**). **C, F** evidenced that the simplified risk score had moderate to excellent performance in classifying tinnitus severity at baseline. **D, E, G, H** showed the evolution of the adjusted risk scores (with CI) (**D, G**), and the performances (Cohen's d and AUC-ROC, (**E, H**) of the model in function of the evolution of tinnitus severity over time. The simplified model had excellent performances to detect individuals at risk of evolving to a severe (**E**) or catastrophic distress (**H**). **C, D, F, G** The boxplots depict the median (center line), interquartile range (box), and 1.5×IQR whiskers. **E, F**, 95% CI estimated across 10,000 bootstrap samples are shown for the effect size.

*answer*. Participants who answered *(4), (6), or (7)* at V1 were excluded. Participants who answered *(6) or (7)* at V2 were excluded. A new category, "No, not now," was constituted for V2 to include participants who answered *(4) Yes, but not now. But have in the past*, and *(5) No. never* to include possible recoveries.

All participants who reported experiencing tinnitus were asked: "How much do these noises worry, annoy, or upset you when they are at their worst?". Possible answers were: *(1) Severely, (2) Moderately, (3) Slightly, (4) Not at all, (5) Do not know, (6) Prefer not to answer*. Participants answering *(5) or (6)* at V1 or V2 were excluded from the analysis.

**Feature selection.** The features were selected a priori by consensus among all authors, based on their relevance to tinnitus as established in the literature[11]. This approach aimed to cover the main domains associated with tinnitus: hearing health, sociodemographic factors, physical health, and mental health. There was no patient or public involvement (PPI) in this selection process. We selected 101 features based on their relevance to tinnitus (more details in supplementary Table S4). Variables were organized into eleven categories forming four distinct domains, as follows:

**Hearing health.** one category with the items: speech-in-noise hearing test, self-reported deafness, self-reported hearing difficulties with or without noise, medical devices (hearing aid or cochlear implants), and noise exposure.

**Mood.** includes three categories (1) neuroticism, based on 12 neurotic behaviors such as irritability, nervous and guilty feelings; (2) traumas (illness, injury, bereavement or stress in the last 2 years); and (3) mood (reported frequency of certain moods in the past 2 weeks and visits to a GP or psychiatrist for nerves, anxiety, tension or depression).

**Physical health.** includes four categories (1) physical activity based on the Metabolic Equivalent Task scores computed using the International Physical Activity questionnaire (IPAQ)[38]; (2) sleep; (3) substance use (smoking and alcohol); and (4) anthropometric measures such as BMI, fractures and blood pressure.

**Sociodemographic.** includes three categories (1) socio-economic status, such as education, income, and employment; (2) occupational measures, such as social entourage and manual or physical job; and (3) demographics, such as age, sex, and ethnicity. Sex is sex defined at birth[39], taken from participants' medical files. Participants had the option to modify this information if needed.

**Missing data.** Since a hearing evaluation was added to V1 a few years after the initial data collection began, if was not performed for 502,237 participants included in the UK Biobank. We included in this study the 192,993 participants who had a hearing evaluation in at least one of the visits and who did not have more than 20% of missing data for the 101 predictors (see Supplementary Fig. S7). For the others, missing data were replaced by the feature median. We verified that the median-only imputation method produced results congruent with a more sophisticated pattern-matching approach for imputing missing data, specifically a data-driven Bayesian ridge regression model (see Supplementary Fig. S8). Features were standardized across participants by centering the mean to zero and scaling the variance to one.

**Data analyses in the UK Biobank**

**Developing the predictive models of tinnitus presence and tinnitus severity.** To identify the risk factors of tinnitus presence and severity, we used the NIPALS regression algorithm (implemented using scikit-learn.org/). This method is especially relevant for high-dimensional clinical datasets as it effectively handles factor multicollinearity. Additionally, it produces interpretable components and offers superior computational efficiency compared to other machine learning algorithms[40]. We used the NIPALS regression algorithm on the 101 features to create risk scores predicting separately (1) Tinnitus presence and (2) Tinnitus associated severity. NIPALS identifies latent patterns that maximize the covariance between two matrices (details in Supplementary Note S1). To this end, the UK Biobank dataset at V1 was divided into a training set ($n = 166,119$ for the presence model, $n = 35,942$ for the severity model) for discovery and a testing set composed of out-of-sample participants for whom longitudinal data were available ($n = 26,874$ for the presence model, $n = 5100$ for the severity model). The algorithms were trained using tenfold cross-validation to estimate the models. The trained models were then applied to the participants of the testing set. The two models' output provided a single prediction for tinnitus presence and its associated severity separately, for each participant. These outputs are referred to as the risk score for tinnitus presence and the risk score for tinnitus severity.

**Tinnitus evolution over time.** The prognostic values of the tinnitus presence and severity risk scores to predict the recovery, persistence, or worsening of tinnitus were assessed using the testing datasets. We created adjusted risk scores, which were orthogonal to the baseline tinnitus level, to interpret if interindividual deviations predicted the evolution over time (Supplementary Fig. S5).

**Tinnitus severity evolution simplified risk score.** A simplified model of the tinnitus severity risk score was derived from the full risk model using the training dataset. Non-modifiable factors (sex, age, and ethnicity), quantitative measures (hand grip strength, hearing test), and composite scores (total neuroticism score, number of life stressors) were excluded from this simplified model to include only modifiable, easily collectable declarative items. We trained a linear forward feature selection algorithm, implemented using scikit-learn, to select the core features that captured the highest explained variance. Features are iteratively added to the model for a prespecified combination of features in the 101 features pool until there is no improvement in the model's performance. We used the elbow method to determine the number of features providing the best trade-off between sparsity and variance explained.

**Table 2 | Baseline characteristics of participants of the Tinnitus Research Initiative**

| | Tinnitus severity (*N* = 463) | | | | | |
|---|---|---|---|---|---|---|
| | **No distress** | **Mild distress** | **Moderate distress** | **Severe distress** | **Catastrophic impact** | ***p* value** |
| Prevalence | 9.3% | 25.9% | 27.9% | 20.5% | 13.8% | |
| Demographics | | | | | | |
| Sex (% M/F) | 67.5/32.6 | 60.0/40.0 | 58.9/41.1 | 64.2/35.8 | 62.5/37.5 | n.s. |
| Age | 57.7 (12.3) | 61.5 (12.8) | 60.2 (11.5) | 60.8 (11.8) | 59.8 (10.3) | n.s. |
| Hearing health | | | | | | |
| Self-reported hearing difficulties | 50.0% | 59.9% | 60.0% | 61.5% | 71.4% | 0.047 |
| Hearing aid users | 17.1% | 18.8% | 18.1% | 61.5% | 71.4% | n.s. |

Data are percentages, except for the Age and Speech reception Threshold for which the mean and standard deviation are given. Group differences were tested with a Chi$^2$ test for the sex proportions, and with an ANOVA for the Age and hearing health features.

### TRI database

The TRI database consists of questionnaire records from patients who visited a tertiary tinnitus clinic[41]. For this study, we included data from 463 individuals out of the 4246 who attended the Interdisciplinary Tinnitus Clinic at the University of Regensburg (Germany). The patients came for their initial visits between May 2004 and February 2022, and their follow-up visits between August 2008 and July 2022, with a median time of four years between visits. These questionnaires assessed participants' tinnitus characteristics, along with aspects of their physical, hearing, and mental health. Further details about the participants are provided in Table 2. We used the Tinnitus Handicap Inventory[42] scores and categorization to determine participants tinnitus severity (no, mild, moderate, severe, catastrophic). An equivalent to the Tinnitus severity evolution simplified risk score was constructed using questions of the TRI database; to determine both the classification and longitudinal validity of the score (Supplementary Note S2).

**Statistical analysis.** The models fit were assessed using the explained variance ($R^2$). The risk scores of individuals with different levels of tinnitus presence or severity were compared to the score of tinnitus-free participants for the presence, and distress-free tinnitus participants for the severity, using Cohen's *d* effect sizes and AUC-ROCs. We used bootstrap resampling with 10,000 iterations to indicate the estimated error in the Cohen's *d* effect sizes. Analyses were performed using Python v.3.11.5 with Spyder 5.4.3, including Numpy (v.1.24.3), Pandas (v.2.0.3), Sklearn (v.1.3.0), Seaborn (v.0.12.2), Matplotlib (v.3.7.2), Pingouin (v.0.5.3), and Nltools (v.0.5.0).

**Ethical approval.** The UKB was approved by the Research Ethics Committee (no. 11/NW/0382). Ethical approval for the collection of the TRI database was obtained from the Ethics Committee of the University of Regensburg (protocol number 08/046). Protocols, consent forms and study procedures were approved by McGill Institutional Review Board. This study received ethics approval under IRB application number: A03-M20-21B (21-03-079). All participants gave written informed consent. The UKB and TRI participants' data were obtained according to the terms and conditions of the databases. The study was not registered.

### Reporting summary

Further information on research design is available in the Nature Portfolio Reporting Summary linked to this article.

### Data availability

The dataset from the UK Biobank analyzed in the study is available via application to the Access Management System at https://www.ukbiobank.ac.uk. The TRI dataset is accessible upon request, see https://tinnitusresearch.net/index.php/for-researchers/tinnitus-database.

### Code availability

Detailed codes and annotations are available at GitHub (https://github.com/EVPlab)[43].

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

## Acknowledgements

L.H. was funded by the European Union's Horizon Europe Framework Program (HORIZON) under the Marie Skłodowska-Curie Postdoctoral Fellowship (grant No. 101146406) and the Fondation des Gueules Cassées. L.H. and S.S. were supported by the Fondation pour l'Audition (FPA RD-2019-10). This work has benefited from a French government grant managed by the Agence Nationale de la Recherche under the France 2030 program, reference ANR-23-IAHU-0003. M.F. was funded by the Louise and Alan Edwards Foundation. CTS was funded by the Canadian Institutes of Health Research (CIHR), the Institut TransMedTech, and the Canada First Research Excellence Fund. E.V.P. was funded by CIHR (CIHR 453096).

## Author contributions

L.H., M.F., M.R., A.L., S.S., and E.V.P. conceived and designed the study. L.H., M.F., and C.T.S. created the methodology. L.H. and M.F. performed the statistical analyses. L.H. drafted the manuscript. E.V.P. supervised the study. All authors were involved in the interpretation of data and the critical revision of the manuscript. All authors had full access to all the data.

## Competing interests

The authors declare no competing interests.
