## [Transparent Peer Review file · Nature Communications]

Tinnitus risk factors and its evolution over time

Corresponding Author: Dr Lise Hobeika

Version 0:

Reviewer comments:

Reviewer #1

(Remarks to the Author)

This study is interesting and used large UK biobank data and another database, Tinnitus Research Initiative, with 463 individuals. Using the UK Biobank dataset, which encompasses data on the socio-demographic, physical, mental and hearing health of more than 170,000 participants, they trained two distinct machine learning models to identify risk scores predicting tinnitus presence and severity separately. These models were used to predict Tinnitus over time and were validated in 463 individuals from the Tinnitus Research Initiative database.

This study seeks to identify socio-demographic, psychological, and health-related risk factors predicting tinnitus presence (how often individuals perceive Tinnitus) and severity separately, as well as their evolution over time. This study aimed to identify the risk factors of tinnitus presence and severity, as well as their evolution over time. The results revealed a dissociation between the features predicting tinnitus presence and tinnitus severity.

There are a lot of machine learning algorithms; this study did not develop any novel algorithm for the analysis. This study used two existing NIPALS regression models for the analysis of the data, but there is no clear explanation/justification for why they used these two regression models.

This study used 101 features to create risk scores. I believe they should apply feature ranking and feature selection models to select the associated and significant features.

In this study, missing data were replaced by the feature median, but in this type of study, there are a lot of missing values, so it is not appropriate to use the feature median to replace the missing values. There are many robust methods, and I suggest applying different other methods that are based on the pattern-matching approach.

Table 1 represents baseline characteristics, including percentages, except for the age and speech reception threshold. But it is possible to put the statistical calculation result where required. It could be difference, odds ratio and many other statistics.

The figures are not sufficiently clear enough to be presented to people who are less familiar with them.

Overall, this study is good but could be expanded by doing additional analysis to identify the pattern of the risk and causal risk factors that affect tinnitus.

Reviewer #2

(Remarks to the Author)

Hobeika et al. perform a longitudinal study on the UKBB dataset to identify factors related to the development of constant or severe tinnitus, and validate them in an independent database of smaller size, but clinical. Authors developed a 5-item questionnaire to identify individuals at risk for developing severe tinnitus and that shows good performance.

This is a long waited study that addresses a major issue in the field, and provides a significant advance. The paper is very well written (with the exception of the introduction, half of which are results), follows a logical analytical flow, the majority of the statistics are sound.

Some numbers are overselling the results giving a false impression on the size of the sample analysed, and the use of only

one follow-up is a weakness. Given the influence of sex on severity, sex stratified analyses are needed (whether as supplemental or in the main ms). Another weakness is the statement that constant tinnitus is assessed, while the definitions of tinnitus in the UKBB data do not allow such precision – the top frequency of tinnitus being “most OR all of the time”. A section on the limitations of the study is needed in order to provide a fair view of the value of these important findings. Codes of the analysis need to be shared for reproducibility purposes.

With hopes these aspects can be addressed, then the manuscript would reach the expected quality.

Major comments:

1. Authors appear to have missed a longitudinal study published by Edvall et al. JCI, 2022 that shows the increasing risk of developing constant tinnitus with increased occurrence of occasional tinnitus (20'000 individuals assessed over 10 years every two years). This study is important because authors observe the important fluctuation of occasional tinnitus states prior becoming constant, something which is known in the field, but for which numbers were lacking.

Due to this variability, Edvall et al. found that cox proportional hazards regression are not appropriate. Instead they used a GEE statistical model to perform the analysis, but using each individuals for which a follow-up was available 2 years after, whenever it was collected over the 10 years period, and even if some individuals were repeated (for instance in V1 and in V4) – the total number of observations is what mattered.

Hobeika et al. would need to

- i) use samples from all other waves to increase the power of their analysis as in Edvall et al., in particular since severe tinnitus is rare (hence, it lacking in the Edvall study) – meaning one time point and one follow-up, even for the same individuals measured there after. If an individual has been measured in V1, V2, V4; then authors should consider V1 -> V2; and V2 -> V4.
- ii) provide ORs in the same way (in order to compare their results with the Edvall study), knowing that the UKBB does not have an “only constant” tinnitus definition.
- iii) Provide a description of the transition of tinnitus (constant or severity) across all possible waves to understand the dynamics of occasional tinnitus with time.

2. More relevant to tinnitus management, but yet completely lacking in the field, is an analysis of the risk to develop both constant AND severe tinnitus. I understand this analysis is difficult to perform as no single question can address this at once. However, Hobeika et al. should perform an analysis of this population, using severe-nonconstant, nonsevere-constant, nonsevere-nonconstant as referent groups; and merging follow-ups as in point 1.i

3. The impact of sex on severity is debated, but the prevalence estimates here go in favor of such notions. Authors should perform sex-stratified analysis for all results presented and show them in the supplemental. How sex was defined in the UKBB also needs to be clear in the methods (self-report, DNA, sex defined at birth, other?), What were the options? was intersex an option?

a. See SAGER guidelines, and refer to them.

4. How features were selected is missing – was this based on consensus across all authors? Was a two round selection process done (e.g. Delphi?)? was the team heterogeneous in expertise? Were patients involved in the selection of the features? If no patient and public involvement (PPI) was used, please mention in methods.

5. A part of the discussion should debate on why the findings from Hobeika et al. where the presence model does not predict tinnitus over time, but severity does, differs from the results reported in Edvall et al. The possibility that the UKBB does not have a good definition of constant only (all the time) is a potential reason of this discrepancy.

6. Descriptives of the TRI database is missing. It is not solely German, it contains many other sources of data. Were are the subset of individuals coming from? Sex, age, socioeconomics etc.. this is completely lacking and is important to consider “validation” of the UKBB data. Hopefully, some features are the same between the two.

7. Section on limitations: Limitations in the study is lacking and should discuss the following

- a. Lack of pure constant tinnitus missing
- b. Recall bias (which would probably be observed when analyzing multiple waves), fluctuation of occasional tinnitus status across waves.
- c. Lack of information on hyperacusis, which is a strong confounder in severe tinnitus (doi: 10.3390/jcm9082412).
- d. Lack of test-retest on a single time point in the newly developed questionnaire in a patient population.

8. The discussion should also consider a discussion with respect of the genetics of constant tinnitus (common variants, see Clifford et al. Nat Comms, 2024) and severe tinnitus (Amanat et al. EBiomedicine, 2022) as a potential non-environmental risk to be combined to the identified factors in the present ms. As a phenotype is the result of genes times the environment, it is important to be shown here, in addition to the fact that some of the risk factors found here, were also identified in the GWAS as genetic correlates to tinnitus.

9. Discussion should also include a discussion on the transition from occasional to constant tinnitus and how this may impact electrophysiological biomarkers like suggested in Edvall et al.

Minor comments:

- Line 44: the paper from Jarach et al. in JAMA Neurol is a global estimate: 14.4% for any tinnitus, and 2.3% for severe tinnitus. Please remove citations 3, 4, 5.
- Line 47: emerging treatments are on the way, and the Lenire trial recently published in Nat. Comms should be mentioned. This is important because risk factors involved in the development of some specific tinnitus subtypes could also be used to stratify individuals into specific emerging treatments.
- Remove the third paragraph of the intro: these are results.
- Provide tables of the numbers of individuals at evaluation and follow-up for figures 1D, 2D.
- Table 1 is misleading. N for tinnitus presence should be 18,615 and for severity 4,291 (line 413). Provide numbers and % together for all possible variables so reviewers can verify the table is correct.
- o The number in Line 274 seems wrong (says 20,850 individuals instead of 18,615)
- o NOTE: these are the numbers of the total population assessed – the number in the abstract makes believe that 170,000 participants were evaluated, which is not the case. See line 27. Please correct to provide the final number evaluated after exclusions and missings.
- o Provide a flowchart of the whole 170,000 individuals and how the numbers go down to the final assessed sample.
- Provide numbers in the methods for any missing data, exclusion so we can track how the authors went from the total UKBB sample down to the evaluated sample. Examples:
- o Line 237, 242, 263, ...
- Provide time range of data acquisition for baseline vs follow-ups.
- What were the variables included as adjustment factors in the risk models?
- What was the statistic program used?
- Line 77: Not all participants that answered the question on presence answered the question on severity. It is thus wrong to state that 20.2% of those with tinnitus experienced moderate or severe distress. In line with the comment above, an additional column with the samples overlapping the two evaluations (presence and severity) would be very valuable.
- Even if data is available from the UKBB, and the TRI, I would assume that the research is performed in France, handling data, and thus an ethics from the French administration would be required. The authors should verify, even post-hoc, whether an ethics would have been necessary, and if not, specify in the methods Line 320, that it wasn't according to local ethics board.
- Figure legend should contain sample size of analyzed groups and subgroups.
- Please harmonize the thickness of bars on graphs from all figures.
- Please provide range of variance in Figure 1 and 2b.
- Please share all cleaned codes and scripts of the analysis e.g. on GitHub. As there are few discrepancies in the numbers, this work will need to be reproduced.
- Please mention if any AI was used for the editing of the manuscript.

Reviewer #3

(Remarks to the Author)

Overall, the manuscript presents valuable insights into tinnitus risk factors and its evolution over time. I appreciate the authors' efforts in establishing a POST five-item questionnaire, which provides a key and practical tool for clinical implication. However, despite its potential clinical impact, this study has several important limitations. For instance, I would suggest the authors to address the weak predictive power of the models, perhaps by adding more features (e.g., genetics, neurobiological data) or further validating the models across more diverse populations. Additionally, I would recommend a deeper discussion of limitations and a more concrete plan for how the POST questionnaire can be integrated into clinical practice. I list some major concerns/questions below:

1. The manuscript distinguishes between "tinnitus" and "tinnitus disorder," but the operationalization of these terms is not sufficiently clear. A clearer, earlier definition in the introduction would help anchor readers. Additionally, consider clarifying the term "phantom sound" and its role in understanding the pathophysiology of tinnitus. This could benefit from a more in-depth discussion about why auditory damage leads to tinnitus in some cases but not others.
2. While the severity model showed promise in predicting tinnitus progression over nine years, the presence model failed to predict tinnitus evolution. This highlights a potential shortcoming of the approach, as predicting who will develop tinnitus should be central to prevention efforts. The manuscript glosses over this issue by shifting focus to severity predictions, but a more critical discussion of why the presence model failed (AUC < 0.60) is warranted. Could it be due to the lack of high-quality audiometric data in the UKB or insufficient consideration of other biological risk factors, such as genetic predispositions?
3. Imputing missing data using feature medians can introduce biases, especially in a dataset as large and heterogeneous as the UKB. This simplistic approach to missing data warrants more critical reflection. Consider either applying more sophisticated imputation methods (e.g., multiple imputation) or providing a sensitivity analysis to demonstrate how different methods of handling missing data might affect the results. This is especially important given the complex interplay between various socio-demographic and health-related factors in the tinnitus models.
4. The identified risk factors, such as mood, neuroticism, and sleep, are consistent with the literature, but they only explain a small proportion of the variance in tinnitus severity (3–7%). This suggests that other important risk factors remain unidentified. The manuscript could benefit from a more critical discussion on this point. For instance, should future studies explore genetic or neurobiological factors in more depth? The brief mention of genetics in the discussion seems insufficient, given the emerging evidence on the genetic contribution to tinnitus.
5. The manuscript acknowledges the lack of ethnic diversity in the UKB sample (91% white). However, this issue needs a more critical exploration, as it limits the generalizability of the findings to non-white populations. In particular, the study

should address whether tinnitus risk factors might differ across ethnic groups, and how future studies could overcome this limitation. Similarly, the differences in tinnitus prevalence and severity between men and women are noted, but further exploration into the gendered nature of these risk factors is missing. Could there be biological or psychosocial mechanisms that explain why men report more tinnitus but women report higher distress?

6. While the POST questionnaire is a useful clinical tool, the manuscript falls short of explaining its real-world applicability. How will this tool be integrated into clinical practice? Will it be used in primary care settings, or is it more suited for specialized clinics? Moreover, there is little discussion of the potential limitations of using a questionnaire-based tool, especially in populations with low health literacy or in clinical settings where tinnitus may be underreported.

Minor comments:

1. Some key references on tinnitus risk factors, particularly from recent meta-analyses and systematic reviews, are missing. For instance, the discussion on hearing health as the main predictor of tinnitus presence could be enriched by including more comprehensive literature reviews. Additionally, consider citing more recent work on the role of stress and mental health in tinnitus severity.

2. Figures, particularly those related to model performance (e.g., ROC-AUC curves), could benefit from clearer annotations. Readers unfamiliar with machine learning or statistical modeling may find the figures challenging to interpret. For instance, the difference between "tinnitus presence" and "tinnitus severity" in the figures could be more explicitly marked. It would also be helpful to label key features driving the models directly on the graphs.

3. Although the manuscript mentions some limitations, such as the dataset's lack of diversity and the inability to predict tinnitus presence evolution, these issues are not explored with enough depth. The manuscript could benefit from a more thorough exploration of why these limitations exist and how future research could address them.

Version 1:

Reviewer comments:

Reviewer #2

(Remarks to the Author)

Authors have adequately addressed my questions and concerns, congratulations for performing the additional analysis, which now provide a very much complete view on tinnitus!

I encourage the authors to consider applying to funds such as the ATA and BTA for implementing a risk calculator using a combination of the data from the UKBB and the new questionnaire to be used by either the public or clinics (see <https://qrisk.org/> - they provide such online tools for cardiovascular risks). This will provide a fantastic advancement for public awareness, risk prevention, and improved care.

Reviewer #4

(Remarks to the Author)

Overall, the authors have addressed the comments with detailed descriptions. They have provided proper justifications for their methodological choices, For instance, their use of median imputation for missing data and the selection of features for the models are now justified properly. Furthermore, they have conducted additional analyses, including sex-stratified evaluations and validations across diverse ethnic groups, which enhanced the generalizability and robustness of their findings.

The authors also mentioned the limitations of their study, including the dataset's lack of diversity, the absence of hyperacusis evaluation, and the limited granularity in tinnitus presence assessment. They provided thoughtful discussions on these points and proposed actionable steps for future research, such as integrating genetic, neurobiological, and other biological factors to improve the predictive power of their models. They have provided an expanded discussion on the applicability of the POST questionnaire in clinical settings, which is crucial for implementing and translating research findings.

Moreover, the authors have revised and enhanced the figures, annotations, and supplemental materials to provide a better understanding of the results. They have sufficiently addressed all the reviewer 3 comments.

Reviewer #5

(Remarks to the Author)

Having carefully read the manuscript, reviewer 1's comments and the authors' response, I believe that the authors have satisfactorily addressed the reviewer's comments.

I think this is a very good paper. The main limitation is the lack of discussion on how to use the tinnitus severity prediction tool in clinical practice, given that its predictive ability is far from optimal and it is not clear what kind of intervention should be applied to subjects at higher risk. This should be more explicitly acknowledged in the limitations of the paper.

Authors Rebuttals to Initial Comments:

We sincerely thank the three reviewers and the editors for their constructive feedback and their positive evaluation of our work. In this revised version of the manuscript, we have placed greater emphasis on identifying risk factors and interpreting the categories of features. Moreover, we performed additional analyses to replicate the findings of Edvall et al., and provided evidence demonstrating the consistencies between the two studies. Additionally, we have improved the manuscript by offering a more detailed explanation of the analyses conducted, a clearer justification for the choice of algorithm, and a more comprehensive description of the external validation cohort beyond the UK Biobank. We also discussed the potential clinical implication of our score and added a limitation sections to increase transparency about the limits of our study. We hope you will find our revisions satisfactory.

Reviewer #1 (Remarks to the Author):

This study is interesting and used large UK biobank data and another database, Tinnitus Research Initiative, with 463 individuals. Using the UK Biobank dataset, which encompasses data on the socio-demographic, physical, mental and hearing health of more than 170,000 participants, they trained two distinct machine learning models to identify risk scores predicting tinnitus presence and severity separately. These models were used to predict Tinnitus over time and were validated in 463 individuals from the Tinnitus Research Initiative database. This study seeks to identify socio-demographic, psychological, and health-related risk factors predicting tinnitus presence (how often individuals perceive Tinnitus) and severity separately, as well as their evolution over time. This study aimed to identify the risk factors of tinnitus presence and severity, as well as their evolution over time. The results revealed a dissociation between the features predicting tinnitus presence and tinnitus severity.

1. There are a lot of machine learning algorithms; this study did not develop any novel algorithm for the analysis. This study used two existing NIPALS regression models for the analysis of the data, but there is no clear explanation/justification for why they used these two regression models.

Response: NIPALS was prioritize over alternative machine learning models because of its robustness in handling multicollinearity. NIPALS captures core data patterns by constructing principal components that represent the variance structure of highly correlated features, which simplifies interpretation without requiring feature selection. We have better justified our choice of machine learning algorithm as follow, Methods section p.8:

“To identify the risk factors of tinnitus presence and severity, we used the NIPALS regression algorithm. This method is especially relevant for high-dimensional clinical datasets as it effectively handles factors multicollinearity. Additionally, it produces interpretable components and offers superior computational efficiency compared to other machine learning algorithms⁴⁰.”

2. This study used 101 features to create risk scores. I believe they should apply feature ranking and feature selection models to select the associated and significant features.

Response: We appreciate the reviewer’s feedback. In this revised version of the manuscript, we have placed greater emphasis on the importance of the features selected by the models. Specifically, we now present the items and their respective weights for predicting the presence of tinnitus (Figure 1A) and tinnitus severity (Figure 2A). Additionally, the structure coefficients (loadings) are provided in the supplementary material (Figures S1). Weights reflect the multivariate pattern learned by the model to predict tinnitus, while loadings illustrate the univariate associations between each feature and the model’s overall prediction. This distinction offers complementary insights: the weights highlight the combined predictive importance of features, whereas the loadings provide a feature-by-feature understanding of their individual associations with the outcome, highlighting the underlying pattern of risk factors for tinnitus.

We also evaluated the significance of each feature category for predicting tinnitus presence and severity (separately) by retraining the model after systematically removing selected features and assessing the resulting impact on overall model performance. These new analyses, now presented in Figure 1D and Figure 2D, highlight the critical role of hearing health in predicting the presence of tinnitus. Specifically, removing hearing-related features reduced model performance significantly, with ROC-AUC dropping from 0.80 to 0.67 in the constant tinnitus group. In contrast, hearing health features were less crucial for predicting tinnitus severity, as their removal only slightly decreased model performance from 0.78 to 0.75 (ROC-AUC in the severe tinnitus group). Thus, the classification of tinnitus severity relied on diverse categories of features, including mood, neuroticism, and sleep problems. This finding underscores the importance of adopting a holistic approach that considers psychosocial factors, beyond hearing health, when identifying risk factors for tinnitus severity. We added a description of those new results in the results section:

Page 3: ‘Retraining the model after removing each category of features showed that features from hearing health were the most important for the prediction of tinnitus presence (Figure 1D)’

Page 3: ‘Retraining the model after removing each category of features showed the multifactorial nature of tinnitus severity, as not a single category was essential to the model performance. Thus, the model accuracy decreased only after removing the four identified categories (hearing, mood, neuroticism and sleep), evidencing the importance of a holistic model for the prediction of tinnitus severity (Figure 2D).’

Lastly, we implemented a feature-ranking approach to develop a simplified version of our risk-factor model for predicting tinnitus severity (POST). Using the elbow method, we determined the optimal number of features (ranging from 1 to 101) that provided the best trade-off between sparsity and explained variance. This analysis identified the top six features that, when combined, offered the most effective balance for predicting tinnitus severity.

In summary, we employed several steps to identify and interpret the pattern of risk factors influencing tinnitus. These included refining weights and loadings, assessing the importance of features through removal experiments, and iteratively ranking features to optimize the POST - a concise six-item questionnaire for predicting tinnitus severity.

3. In this study, missing data were replaced by the feature median, but in this type of study, there are a lot of missing values, so it is not appropriate to use the feature median to replace the missing values. There are many robust methods, and I suggest applying different other methods that are based on the pattern-matching approach.

*Response: We thank the reviewer for this suggestion. Our decision to use median imputation was informed by previous work conducted by our group with the UK Biobank, where we compared Bayesian imputation and simple median imputation for psychosocial variables predicting chronic pain. This comparison showed minimal differences in downstream machine learning predictive performance when predicting chronic pain (see Tanguay-Sabourin, et al. "A prognostic risk score for development and spread of chronic pain." *Nature Medicine* 29.7 (2023): 1821-1831.). Here, we confirmed that Bayesian and median imputation yielded similar results for our tinnitus presence and severity models. The figure below shows the comparison between median and Bayesian imputation methods. The model performance and the weights of features remained the same regardless of the method of imputation. We have updated the manuscript to include these results (supplementary Figure S8) and a discussion on our rationale for the chosen imputation method is provided page 8.*

“Missing data were replaced by the feature median. We verified that the median-only imputation method produced results congruent with a more sophisticated pattern-matching approach for imputing missing data, specifically a data-driven Bayesian ridge regression model (see supplementary Figure S8).”

Data imputation
Comparison between different methods

A

Presence Model

ROC/AIC	Some of the time	A bit of the time	Most or all the time
Median Imputation	.68	.75	.80
Bayesian Imputation	.68	.75	.80

B

Severity Model

ROC/AIC	Mild	Moderate	Severe
Median Imputation	.52	.74	.79
Bayesian Imputation	.51	.74	.79

Model weights, in function of the data imputation method

Model weights, in function of the data imputation method

4. Table 1 represents baseline characteristics, including percentages, except for the age and speech reception threshold. But it is possible to put the statistical calculation result where required. It could be difference, odd ratio and many other statistics.

Response: We added statistics in Table 1 and information in the results section.

	Tinnitus Presence (N = 192,993)				Tinnitus Severity (N = 41,042)					pvalue
	Never had tinnitus	Tinnitus some of the time	Tinnitus a lot of the time	Tinnitus all the time	pvalue	No distress	Mild distress	Moderate distress	Severe distress	
Prevalence	78.7 %	10.4 %	3.0 %	7.9 %		29.0 %	48.3 %	19.0 %	3.7 %	
Demographics										
Sex (% M/F)	44.2 / 55.8	48.6 / 51.4	53.4 / 46.6	59.5 / 40.5	<10 ⁻³⁰⁰	55.9 / 44.1	48.8 / 51.2	46.9 / 53.1	44.4 / 55.6	10 ⁻⁷⁸
Age	58.1 (87)	59.7 (8.3)	60.8 (7.8)	61.4 (7.3)	<10 ⁻³⁰⁰	58.9 (6.6)	59.2 (6.2)	59.1 (6.5)	58.5 (6.8)	n.s.
Hearing Health										
Self-reported hearing difficulties	20.1 %	42.6 %	55.8 %	67.9 %	<10 ⁻³⁰⁰	43.9 %	53.2 %	67.4 %	76.7 %	<10 ⁻³⁰⁰
Self-reported speech-in-noise hearing difficulties	31.0 %	52.3 %	61.2 %	67.2 %	<10 ⁻³⁰⁰	51.8 %	61.1 %	71.9 %	78.5 %	10 ⁻²²³
Speech Reception Threshold in dB	-6.4 (1.8)	-6.1 (2.1)	-5.9 (2.1)	-5.9 (2.2)	<10 ⁻³⁰⁰	-6.2 (1.9)	-6.1 (1.9)	-5.9 (2.2)	-5.5 (2.7)	10 ⁻⁵⁰
Hearing aid users	2.6 %	6.6 %	8.5 %	14.1 %	<10 ⁻³⁰⁰	8.0 %	8.6 %	13.2 %	19.8 %	10 ⁻⁵⁹
Cochlear implants	.054 %	.25 %	.17 %	.22 %	<10 ⁻¹⁸	.18 %	.19 %	.32 %	.72 %	10 ⁻³

5. The figures are not sufficiently clear enough to be presented to people who are less familiar with them.

Response: We have made several improvements to enhance the clarity of the figures for readers who may be less familiar with the subject. Specifically, we have added detailed annotations to display the features used to train the model and expanded the figure legends to provide more comprehensive descriptions of the results in each panel. These changes aim to ensure that the figures are more accessible and informative for all audiences.

6. Overall, this study is good but could be expanded by doing additional analysis to identify the pattern of the risk and causal risk factors that affect tinnitus.

Response: We appreciate the reviewer's positive feedback on our study. In response to their query, we have conducted additional analyses, which are detailed in Point #2 above. Briefly, we now present the features and their corresponding weights in Figures 2 and 3 of the manuscript, representing the multivariate patterns that best predict tinnitus. To enhance the interpretability of the models, we also

include the loadings of the features, which represent the univariate association between each feature and the model's prediction. Furthermore, we evaluated the importance of feature categories by systematically removing them from the model and assessing the impact on its performance. Finally, we developed a simplified model optimized for prediction using only the top-ranked features. This process identified a six-item questionnaire, the POST, for predicting tinnitus severity. The generalizability of the POST was confirmed by demonstrating its equivalent performance on an independent dataset.

Reviewer #2 (Remarks to the Author):

Hobeika et al. perform a longitudinal study on the UKBB dataset to identify factors related to the development of constant or severe tinnitus, and validate them in an independent database of smaller size, but clinical. Authors developed a 5-item questionnaire to identify individuals at risk for developing severe tinnitus and that shows good performance.

This is a long-awaited study that addresses a major issue in the field, and provides a significant advance. The paper is very well written (with the exception of the introduction, half of which are results), follows a logical analytical flow, the majority of the statistics are sound.

Response: We are thankful for the kind feedback.

Some numbers are overselling the results giving a false impression on the size of the sample analysed, and the use of only one follow-up is a weakness.

Response: The sample size varied depending on the type of analysis performed. For analyses related to the presence of tinnitus, a total of 192,993 participants were included. Of these, 166,119 participants were used to train the models, while 26,874 were reserved for testing the model's performance on out-of-sample data. For tinnitus severity, the reviewer is correct that the sample size was smaller. In this case, approximately 41,042 participants were included in the analyses. Of these, 35,942 participants were used to train the models and determine the optimal weighting for predicting tinnitus severity, while 5,100 participants were held out for validation purposes.

The apparent 'overselling' argument regarding the number of participants may stem from the inclusion of a large group of participants (approximately 150,000) who did not report tinnitus. It is important to clarify that participants without tinnitus were incorporated into our analyses, enabling the model to discern differences between individuals without tinnitus and those experiencing varying levels of tinnitus. Specifically, the NIPALS algorithm was trained to predict four distinct levels of tinnitus presence: 1 (no tinnitus), 2 (tinnitus some of the time), 3 (tinnitus a lot of the time), and 4 (tinnitus most or all of the time). After training the model, it was applied to a separate validation cohort to assess its ability to differentiate individuals with varying degrees of tinnitus from healthy controls (results shown in figures 1 and 2). The no tinnitus group were important for learning what characterized (or not) the presence of tinnitus. In summary, our machine learning models were trained on over 190,000 participants for tinnitus presence and over 40,000 participants for tinnitus severity. We believe these substantial sample sizes make this study one of the largest investigations of its kind. The number of participants with tinnitus are however now more clearly reported in a figure flowchart presented below (see minor points) and are now explicitly reported in the abstract of the study.

Given the influence of sex on severity, sex stratified analyses are needed (whether as supplemental or in the main ms).

Response: Sex stratified analyses have now been performed and are presented below in point #3.

Another weakness is the statement that constant tinnitus is assessed, while the definitions of tinnitus in the UKBB data do not allow such precision – the top frequency of tinnitus being “most OR all of the time”. A section on the limitations of the study is needed in order to provide a fair view of the value of these important findings.

Response: The reviewer is correct, and we discussed this point in the limitation section at the end of the discussion (see point #7).

Codes of the analysis need to be shared for reproducibility purposes.

Response: All the analysis codes of this manuscript are on Github: <https://github.com/EVPlab>.

With hopes these aspects can be addressed, then the manuscript would reach the expected quality.

Major comments:

1. Authors appear to have missed a longitudinal study published by Edvall et al. JCI, 2022 that shows the increasing risk of developing constant tinnitus with increased occurrence of occasional tinnitus (20'000 individuals assessed over 10 years every two years). This study is important because authors observe the important fluctuation of occasional tinnitus states prior becoming constant, something which is known in the field, but for which numbers were lacking. Due to this variability, Edvall et al. found that cox proportional hazards regression are not appropriate. Instead, they used a GEE statistical model to perform the analysis, but using each individual for which a follow-up was available 2 years after, whenever it was collected over the 10 years period, and even if some individuals were repeated (for instance in V1 and in V4) – the total number of observations is what mattered. Hobeika et al. would need to

Response: We thank the reviewer for pointing out the Edvall study that is relevant to our manuscript. The study is now cited three times in our manuscript as follow:

page 2: 'This deafferentation disrupts auditory inputs along the auditory pathway, prompting the nervous system to generate the perception of sound in the absence of an external stimulus'⁹

page 5: 'the prevalence of tinnitus and the odds ratio for its evolution are aligned with the literature.'^{2,9,17-19}

page 6: 'Additionally, evidence from neuroimaging studies points to distinct patterns of brain functioning associated with tinnitus^{36,37}. Edvall et al. recently demonstrated that altered auditory brainstem responses distinguish constant tinnitus from occasional tinnitus⁹. These findings underscore the importance of incorporating a broader range of environmental and biological factors in future studies.'

i) use samples from all other waves to increase the power of their analysis as in Edvall et al., in particular since severe tinnitus is rare (hence, it lacking in the Edvall study) – meaning one time point and one follow-up, even for the same individuals measured thereafter. If an individual has been measured in V1, V2, V4; then authors should consider V1->V2; and V2 -> V4.

Response: We would like to emphasize that our analyses already included over 160,000 participants for the analysis of tinnitus presence and over 40,000 for tinnitus severity. To ensure that our models were not overfitted due to power failure, we employed cross-validation and evaluated model performance on an independent holdout test set. Additionally, following best practices in machine learning, we assessed the generalizability of our models using an external cohort of new participants, as recommended by leading methodologies (e.g., Poldrack, R. A., Huckins, G., & Varoquaux, G. (2020). Establishment of best practices for evidence for prediction: a review. *JAMA psychiatry*, 77(5), 534-540.).

Yet, the reviewer's suggestion is well taken, and we further expanded the sample size by incorporating data from all evaluation waves, rather than limiting it to waves T0 and T2 as in the original analysis. Previously, participants who did not undergo a hearing evaluation at the first visit (T0) were excluded, even if they were assessed during later visits (T1, T2, or T3). Moreover, we have now excluded at baseline the participants who used to have tinnitus, but no longer have it, in the severity analysis. By incorporating these changes, we increased the size of both the presence dataset and the severity dataset (see Supplementary Figure S7 and Table S3 for a full description of the updated datasets).

Here, changing the sample did not modify the model for presence prediction but slightly improved the accuracy of the severity model. We have now updated the manuscript to reflect these new analyses, which led to minor changes in the features selected by the algorithm and, consequently, in the POST

questionnaire (now including 6 questions instead of 5). The primary domains represented in the questionnaire—hearing health, mental health, and sleep disorders—remain the same, although one selected mental health item have changed, and an additional mental health item was included. This adjustment enhanced the performance of our model applied in the clinical cohort outside the UKBiobank. We are therefore thankful to the reviewer for their suggestions.

ii) provide ORs in the same way (in order to compare their results with the Edvall study), knowing that the UKBB does not have an “only constant” tinnitus definition.

Response: We would like to emphasize that our study was fundamentally different in scope and objective than the Edvall’s study. One of the primary goal of the Edvall study was to characterize the prevalence of tinnitus over repeated visits, demonstrating that individuals with constant tinnitus were more likely to experience persistent tinnitus two years later, while those with intermittent tinnitus exhibited more variable outcomes. In contrast, our study focuses on identifying a multivariate pattern of risk factors that classified different levels of tinnitus (presence and severity separately). Tinnitus was kept as the target we are trying to predict using a series of tinnitus agnostic measurements, including hearing health, mood, sleep, socioeconomic factors amongst other.

The reviewer’s point is however well taken and we calculated the odds ratios (ORs) for tinnitus progression over time. The matrix below provides an analysis of ORs across all levels of tinnitus presence and severity, showing that individuals with most or all the time tinnitus were more likely to experience stable tinnitus at follow up visits. Our findings showed consistent results with the Edvall study, when accounting for differences in follow-up duration (2 years in Edvall et al. versus 9 years in our study) and the categorization of tinnitus presence. We have included these ORs in supplementary analysis Figure S4 and reported that these findings validate and solidifies the conclusions of the Edvall study.

The discussion now reports in page 5 that: ‘the prevalence of tinnitus and the odds ratio for its evolution are aligned with the literature for this age range.^{2,9,17–19,}

iii) Provide a description of the transition of tinnitus (constant or severity) across all possible waves to understand the dynamics of occasional tinnitus with time.

Response: The transitions between tinnitus status are represented in the matrix below, where the heatmaps depict the percentage of individuals within each combination of baseline and follow-up status. Here again, the results for tinnitus presence are consistent with those reported in Table S2 of Edvall et al. (2022). While these findings are also interesting, we chose to present the odds ratio presented in point #2 to corroborate the Edvall et al conclusion.

2. More relevant to tinnitus management, but yet completely lacking in the field, is an analysis of the risk to develop both constant AND severe tinnitus. I understand this analysis is difficult to perform as no single question can address this at once. However, Hobeika et al. should perform an analysis of this population, using severe-nonconstant, nonsevere-constant, nonsevere-nonconstant as referent groups; and merging follow-ups as in point 1.i

*Response: We are thankful to the reviewer for their suggestion. We trained and tested three different models to distinguish **Constant Tinnitus with Severe Distress** from:*

- A) **Intermittent Tinnitus without Distress***
- B) **Constant Tinnitus without Distress***
- C) **Intermittent Tinnitus with Severe Distress***

Overall, training these three new models suggest that

- Changes in both tinnitus presence and severity are influenced by hearing health, mood, and neuroticism (Figure below in A).*
- Changes in tinnitus severity are linked to mood, neuroticism (Figure below in B).*
- The third model had a low accuracy (AUC-ROC = .65), probably due to the low sample size, therefore the explained variance of each category was not interpreted.*

These findings align with the models presented in the article and support the initial conclusions of our manuscript.

A

Tinnitus All the time with Severe distress VS Tinnitus some of the time without distress

B

Tinnitus All the time with Severe distress VS Tinnitus All the time without distress

C

Tinnitus All the time with Severe distress VS Tinnitus some of the time with Severe distress

3. The impact of sex on severity is debated, but the prevalence estimates here go in favor of such notions. Authors should perform sex-stratified analysis for all results presented and show them in the supplemental. How sex was defined in the UKBB also needs to be clear in the methods (self-report, DNA, sex defined at birth, other?), What were the options? was intersex an option? a. See SAGER guidelines, and refer to them.

Response: This is an excellent point. Sex as defined in the UKB is sex defined at birth, taken from participants' medical file. Participants had the option to modify this information if needed. Only 0.1% of participants had chromosomal sex types other than XX or XY, which may be indicative of intersex variations. We added in the methods information on the definition of sex in the UKB, citing SAGER guidelines page 8

“Sex is sex defined at birth³⁹, taken from participants’ medical file. Participants had the option to modify this information if needed.”

We performed a sex-stratified analysis (presented in supplementary figure S5) that confirms that the presence and severity models have similar accuracy on both sexes independently.

Sex stratified analysis
Presence model

Cross-sectional Longitudinal

Analysis in Women

Analysis in Men

Sex stratified analysis
Severity model

Cross-sectional Longitudinal

Analysis in Women

Analysis in Men

4. How features were selected is missing – was this based on consensus across all authors? Was a two round selection process done (e.g. Delphi)? was the team heterogeneous in expertise? Were patients involved in the selection of the features? If no patient and public involvement (PPI) was used, please mention in methods.

Response: The features were selected a priori by consensus among all authors, based on their relevance to tinnitus as established in the literature, particularly by the review and meta-analysis presented by Biswas et al. (2023). This approach aimed to cover the main domains associated with tinnitus: hearing health, socio-demographic factors, physical health, and mental health. There was no patient or public involvement (PPI) in this selection process. We added that information in the methods page 7

“The features were selected a priori by consensus among all authors, based on their relevance to tinnitus as established in the literature¹¹. This approach aimed to cover the main domains associated with tinnitus: hearing health, socio-demographic factors, physical health, and mental health. There was no patient or public involvement (PPI) in this selection process.”

5. A part of the discussion should debate on why the findings from Hobeika et al. where the presence model does not predict tinnitus over time, but severity does, differs from the results reported in Edvall et al. The possibility that the UKBB does not have a good definition of constant only (all the time) is a potential reason of this discrepancy.

Response: The conclusion of our study seems rather aligned with Edvall et al. results, though the goals and methodologies of the two studies differ significantly. The key finding in Edvall et al. is the calculation of odds ratios (ORs) for experiencing constant tinnitus based on prior tinnitus status two years earlier among individuals already experiencing tinnitus. Specifically, their study reported that individuals who experienced tinnitus “sometimes” had an OR of 5.62 for developing constant tinnitus after two years, those with tinnitus “often” had an OR of 29.74, and those with constant tinnitus had an OR of 603.02 in subsequent visits. These findings are consistent with ours, particularly when accounting for differences in follow-up duration (2 years in Edvall et al. versus 9 years in our study) and the categorization of tinnitus presence. Similarly, we observed higher ORs for persistent tinnitus (“most or all of the time”) in participants who reported similar symptoms nine years prior compared to those who experienced tinnitus “some of the time” or “a lot of the time” (see Supplementary Figure S8). This finding has been demonstrated earlier in point ii. and iii. and is now included in the revised manuscript.

*We also added in the discussion that the lack of constant tinnitus may be a limitation, page 6:
“Second, the assessment of tinnitus presence lacks granularity in the UK Biobank, as it does not include a pure constant tinnitus category.”*

6. Descriptives of the TRI database is missing. It is not solely German, it contains many other sources of data. Where are the subset of individuals coming from? Sex, age, socioeconomics etc.. this is completely lacking and is important to consider “validation” of the UKBB data. Hopefully, some features are the same between the two.

Response:

The TRI database includes data collected from 11 centers worldwide. The Regensburg center serves as the primary hub, having collected 4,894 observations at the time of this study (with a subset of participants having longitudinal evaluations). The other centers contributed with significantly less data, one site collecting nearly 200 observations and the others between 20 and 90.

While the Regensburg center holds full authorization to share their data with us, this is not the case for the other centers. Using their data would require obtaining independent ethical approval from each center. For this reason, and because we were interested by the longitudinal data to validate our model, we limited our study to participants from the Regensburg center.

We added more information about the participants in the methods page 9 and in Table 2 to depict the demographic information of those patients.

“The TRI database consists of questionnaire records from patients who visited a tertiary tinnitus clinic.³⁸ For this study, we included data from 463 individuals out of the 4,246 who attended the Interdisciplinary Tinnitus Clinic at the University of Regensburg (Germany). The patients came for their initial visits between May 2004 and February 2022, and their follow-up visit between August 2008 and July 2022, with a median time of four years between visits. These questionnaires assessed participants’ tinnitus characteristics, along with aspects of their physical, hearing, and mental health. Further details about the participants are provided in Table 2.’

	Tinnitus Severity (N = 463)					pvalue
	No distress	Mild distress	Moderate distress	Severe distress	Catastrophic impact	
Prevalence	9.3%	25.9%	27.9%	20.5%	13.8%	
Demographics						
Sex (% M/F)	67.5 / 32.6	60.0 / 40.0	58.9 / 41.1	64.2 / 35.8	62.5 / 37.5	n.s.
Age	57.7 (12.3)	61.5 (12.8)	60.2 (11.5)	60.8 (11.8)	59.8 (10.3)	n.s.
Self-reported hearing difficulties	50.0%	59.9%	60.0%	61.5%	71.4%	< .05
Hearing aid users	17.1%	18.8%	18.1%	61.5%	71.4%	n.s.

7. Section on limitations: Limitations in the study is lacking and should discuss the following a. Lack of pure constant tinnitus missing. b. Recall bias (which would probably be observed when analyzing multiple waves), fluctuation of occasional tinnitus status across waves. c. Lack of information on hyperacusis, which is a strong confounder in severe tinnitus (doi: 10.3390/jcm9082412). d. Lack of test-retest on a single time point in the newly developed questionnaire in a patient population.

Response: We are thankful for the suggestion. We added those limitations on the discussion and the one proposed by reviewer 3.

Page 5 “Overall, even if the UK Biobank has potential biases such as healthy volunteer selection bias¹⁶, recall bias, lack of test-retest on a single time point in the newly developed questionnaire,.... ”

Page 6 “Second, the hearing and tinnitus evaluations available in the UK Biobank could be more precise. Specifically, the assessment of tinnitus presence lacks granularity, as it includes a most or all the time item, which may not reflect a pure constant tinnitus category. Additionally, the absence of hyperacusis evaluation - a hypersensitivity to noise commonly co-occurring with tinnitus - limits our ability to account for this important comorbidity³³.”

8. The discussion should also consider a discussion with respect of the genetics of constant tinnitus (common variants, see Clifford et al. Nat Comms, 2024) and severe tinnitus (Amanat et al. EBiomedicine, 2022) as a potential non-environmental risk to be combined to the identified factors in the present ms. As a phenotype is the result of genes times the environment, it is important to be shown here, in addition to the fact that some of the risk factors found here, were also identified in the GWAS as genetic correlates to tinnitus.

Response: We added more information on the genetic contribution on the discussion page 6

“Third, adding important biological factors in our analysis, such as genetic and cerebral contributions^{34,35}, should be considered in the future, especially for predicting presence of tinnitus as recent studies highlight the significant role of genetics. For instance, Clifford et al. identified specific genetic architectures that differentiate the perception of tinnitus from hearing loss³⁴.”

9. Discussion should also include a discussion on the transition from occasional to constant tinnitus and how this may impact electrophysiological biomarkers like suggested in Edvall et al.

Response: We added the following in the discussion page 6

“Additionally, evidence from neuroimaging studies points to distinct patterns of brain functioning associated with tinnitus^{36,37}. Moreover Edvall et al. recently demonstrated that altered auditory brainstem responses distinguish constant tinnitus from occasional tinnitus⁹.”

Minor comments:

• Line 44: the paper from Jarach et al. in JAMA Neurol is a global estimate: 14.4% for any tinnitus, and 2.3% for severe tinnitus. Please remove citations 3, 4, 5.

Response: In addition to Jarach et al. (JAMA Neurol), we retained references 3 and added Batts & Stankovic 2024, which are prevalence studies published in Lancet Regional Health in 2024. Citation 4 and 5 have been removed.

• Line 47: emerging treatments are on the way, and the Lenire trial recently published in Nat. Comms should be mentioned. This is important because risk factors involved in the development of some specific tinnitus subtypes could also be used to stratify individuals into specific emerging treatments.

Response: We added the suggested reference in the introduction

• Remove the third paragraph of the intro: these are results.

Response: We updated this paragraph but still describe the aims, the novelty, and the procedure.

• Provide tables of the numbers of individuals at evaluation and follow-up for figures 1D, 2D.

Response: We added this in the supplementary in Tables S1 and Table S2.

• Table 1 is misleading. N for tinnitus presence should be 18,615 and for severity 4,291 (line 413). Provide numbers and % together for all possible variables so reviewers can verify the table is correct.

Response: The reviewer’s assertion seems incorrect, as explained earlier. The analysis sample sizes (192,993 for the presence model and 44,122 for the severity model) are clearly defined and include both the training and testing datasets. Specifically, the training dataset consists of 166,119 participants for the presence model and 38,805 participants for the severity model, while the testing dataset comprises 26,874 out-of-sample participants for the presence model and 5,137 for the severity model, all of whom had longitudinal data available. It is important to note that participants without tinnitus were also included in our analyses as such in the flowchart. Their inclusion was crucial for the model to learn the distinctions between individuals without tinnitus and those with varying levels of presence. The large sample size is particularly significant as it ensures the study is sufficiently powered, reducing the risk of overfitting and enhancing the model’s generalizability.

- The number in Line 274 seems wrong (says 20,850 individuals instead of 18,615)

Response: We modified it to match the new dataset composition.

- NOTE: these are the numbers of the total population – the number in the abstract makes believe that 170,000 participants were evaluated, which is not the case. See line 27. Please correct to provide the final number evaluated after exclusions and missings.

Response: A total of 192,903 participants had an evaluation of their hearing health, including question about tinnitus. These were all included in the training of our predictive model. Those numbers have been corrected to match the constitution of the datasets after considering all the waves. To increase transparency, we now also reported the number of participants reporting tinnitus in the abstract.

- Provide a flowchart of the whole 170,000 individuals and how the numbers go down to the final assessed sample. • Provide numbers in the methods for any missing data, exclusion so we can track how the authors went from the total UKBB sample down to the evaluated sample. Examples: Line 237, 242, 263, ...

Response: The following flowchart have been included as Figure S7 for maximal transparency.

We also included a new Table S3

Tables S3: Constitution of the dataset of 217,261 participants out of the 502,367 participants of the UK Biobank.

The UK Biobank is constituted of one baseline visit (T0, n = 502,367 participants) and three follow-up visits for which a different subset of participants was included (T1: n = 20,343 participants, T2: n = 65,655 participants and T3: n = 5,357 participants). Hearing evaluation was not included in the dataset at the start of the data collection, thus only a sub-sample of participants at T0 were evaluated for their audition. To add as many participants as possible in our analysis, we added data of all participants who had at least one hearing evaluation in one of the visits. Table A describes the number of participants included in our analysis who had **only one** hearing evaluation. We described the visit of the UK Biobank at which they received this evaluation. If a participant had **at least two** hearing evaluations, we included the longitudinal evaluation in our dataset. Table B describes the number of participants who received at least two hearing evaluation, and at which UK Biobank visits they were performed. When participants had 3 or 4 hearing evaluations, we always kept the older evaluation as baseline, and the one following as a follow-up. For example, if a participant was evaluated at T0, T2 and T3, we included his evaluation at T0 as the baseline evaluation, and T2 as the follow-up evaluation.

Data at T0 were collected between April 2009 and October 2010. Data at T1 were collected between August 2012 and June 2013. Data at T2 were collected between April 2014 and march 2023. Finally, data at T3 were collected between June 2019 and February 2023

A. Number of participants with only one hearing evaluation

Visit	
T0	145,025
T1	8,631
T2	33,544
T3	47

B. Number of participants with at least two hearing evaluations

		Second visit (Follow-up)		
		T1	T2	T3
First visit (Baseline)	T0	4,625	18,568	20
	T1		6,791	10
	T2			0

Participants Flowchart

DATASET OF THE PRESENCE MODEL

DATASET OF THE SEVERITY MODEL

- **Provide time range of data acquisition for baseline vs follow-ups.**

Response: The information was added in the methods, for both datasets.

Page 7: ‘We used data from the all the visits to constitute the baseline (V1, collected between April 2009 and November 2021) and the follow up (V2, collected between August 2012 and February 2023) datasets. More information about the constitution of those datasets is available in the supplementary Figures S1 and Table S1.’

Page 9: ‘For this study, we included data from 463 individuals out of the 4,246 who attended the Interdisciplinary Tinnitus Clinic at the University of Regensburg (Germany). The patients came for their initial visits between May 2004 and February 2022, and their follow-up visit between August 2008 and July 2022, with a median time of four years between visits.’

- **What were the variables included as adjustment factors in the risk models?**

Response: There were no adjustment factors in the models. All variables were included as variables of interest and entered in the model.

- **What was the statistic program used?**

Response: Analyses were performed using Python v.3.11.5 with Spyder 5.4.3, including Numpy (v.1.24.3), Pandas (v.2.0.3), Sklearn (v.1.3.0), Seaborn (v.0.12.2), Matplotlib (v.3.7.2), Pingouin (v.0.5.3) and Nltools (v.0.5.0). This information is available in the method section.

- **Line 77: Not all participants that answered the question on presence answered the question on severity. It is thus wrong to state that 20.2% of those with tinnitus experienced moderate or severe distress. In line with the comment above, an additional column with the samples overlapping the two evaluations (presence and severity) would be very valuable.**

Response: All participants who reported having tinnitus were asked the question about the severity. The numbers can be verified on the flowchart in the supplementary in Figures S1 and Table S1 and the text has been corrected accordingly to those numbers.

- **Even if data is available from the UKBB, and the TRI, I would assume that the research is performed in France, handling data, and thus an ethics from the French administration would be required. The authors should verify, even post-hoc, whether an ethics would have been necessary, and if not, specify in the methods Line 320, that it wasn't according to local ethics board.**

Response: We added this information in the method section page 9. The analyses were performed at McGill University in Canada in the last author lab and the group has ethic approval (Vachon-Presseau).

“Protocols, consent forms and study procedures were approved by McGill Institutional Review Board. This study received ethics approval under IRB application number: A03-M20-21B (21-03-079)”

- **Figure legend should contain sample size of analyzed groups and subgroups.**

Response: We added this precision in the legends.

- **Please harmonize the thickness of bars on graphs from all figures.**

Response: They have now been harmonized.

- **Please provide range of variance in Figure 1 and 2b.**

Response: Figures 1C and 2C depicts the explained variance of each category. There is no range for this analysis.

• **Please share all cleaned codes and scripts of the analysis e.g. on GitHub. As there are few discrepancies in the numbers, this work will need to be reproduced.**

Response: All codes have been uploaded the analysis scripts to GitHub, and this information has been added to the manuscript in the statistical analysis subsection.

• **Please mention if any AI was used for the editing of the manuscript.**

Response: The manuscript was all written by the authors. Grammar was verified using ChatGPT and Grammarly, but no text was generated using AI. Our use of AI is in compliance with the definition of AI assisted copy editing defined by the Nature Portfolio guidelines (<https://www.nature.com/nature-portfolio/editorial-policies/ai>) stating that: ‘The use of an LLM (or other AI-tool) for “AI assisted copy editing” purposes does not need to be declared. In this context, we define the term “AI assisted copy editing” as AI-assisted improvements to human-generated texts for readability and style, and to ensure that the texts are free of errors in grammar, spelling, punctuation and tone. These AI-assisted improvements may include wording and formatting changes to the texts, but do not include generative editorial work and autonomous content creation. In all cases, there must be human accountability for the final version of the text and agreement from the authors that the edits reflect their original work.’

Reviewer #3 (Remarks to the Author):

Overall, the manuscript presents valuable insights into tinnitus risk factors and its evolution over time. I appreciate the authors' efforts in establishing a POST five-item questionnaire, which provides a key and practical tool for clinical implication. However, despite its potential clinical impact, this study has several important limitations. For instance, I would suggest the authors to address the weak predictive power of the models, perhaps by adding more features (e.g., genetics, neurobiological data) or further validating the models across more diverse populations. Additionally, I would recommend a deeper discussion of limitations and a more concrete plan for how the POST questionnaire can be integrated into clinical practice. I list some major concerns/questions below:

Response: We are thankful for the positive feedback. A point-by-point response is detailed below.

1. The manuscript distinguishes between “tinnitus” and “tinnitus disorder,” but the operationalization of these terms is not sufficiently clear. A clearer, earlier definition in the introduction would help anchor readers. Additionally, consider clarifying the term "phantom sound" and its role in understanding the pathophysiology of tinnitus. This could benefit from a more in-depth discussion about why auditory damage leads to tinnitus in some cases but not others.

Response: Thank you for this remark. A large consortium proposed new definitions to distinguish between tinnitus (the perception of tinnitus) and tinnitus disorder (perception of tinnitus with an associated distress). However, the use of this terminology is still debated in the research community (for instance, the alternative definition proposed in Noreña et al., A contribution to the debate on tinnitus definition, Progress in Brain Research, 262, 2021, pp. 469–485). Instead of using this framework in our study, we would rather analyze the presence and severity separately, and confront our results to the definitions. We re-write the discussion without referencing the distinction between tinnitus and tinnitus disorder.

“Overall, our study clearly distinguishes between the presence and severity of tinnitus. Tinnitus presence, associated with hearing health, highlights the necessity for raising public awareness about the irreversible consequences of peripheral auditory damage induced by noise or ototoxic drugs. Conversely, tinnitus severity is influenced by psychosocial factors, underscoring the significance of interventions targeting these factors. It is also influenced by hearing health, suggesting that deeper hearing loss will trigger more intrusive tinnitus. Additionally, we show that only tinnitus severity can be predicted, pinpointing differences in risk factors associated with each dimension.”

We re-write a section of the introduction to have a deeper explanation of the term phantom sound

“Tinnitus is thought to arise from a maladaptive reaction to auditory peripheral damage caused by factors such as presbycusis, noise exposure, ototoxic medication, or trauma, leading to sensory deafferentation. This deafferentation disrupts auditory inputs along the auditory pathway, prompting the brain to generate the perception of sound in the absence of an external stimulus. In this context, tinnitus is considered a phantom sound, analogous to phantom limb pain, where individuals perceive sensations in a missing limb due to maladaptive neural activity in the brain.”

2. While the severity model showed promise in predicting tinnitus progression over nine years, the presence model failed to predict tinnitus evolution. This highlights a potential shortcoming of the approach, as predicting who will develop tinnitus should be central to prevention efforts. The manuscript glosses over this issue by shifting focus to severity predictions, but a more critical discussion of why the presence model failed (AUC < 0.60) is warranted. Could it be due to the lack of high-quality audiometric data in the UKB or insufficient consideration of other biological risk factors, such as genetic predispositions?

Response: Ultimately, the inability to accurately predict tinnitus presence based on hearing health and psychosocial variables remains unclear. One possibility is that tinnitus onset may be time-locked with the development of hearing deficits, and measuring these deficits nine years earlier may lack the precision needed to predict the condition. Another consideration is that incorporating biological factors into the model might enhance its predictive power. Alternatively, the evolution of tinnitus may simply be too unpredictable due to the heterogeneity of its underlying causes or the challenges in forecasting age-related deterioration of hearing health that had not yet manifested at the baseline visit. In all cases, what we do know is that the evolution of the severity of the tinnitus can be accurately predicted nine years in advance and could therefore be targeted to improve the quality of life of the patients. We added a discussion on the limitations page 6

“Third, adding important biological factors in our analysis, such as genetic and cerebral contributions^{34,35}, should be considered in the future, especially for predicting presence of tinnitus as recent studies highlight the significant role of genetics. For instance, Clifford et al. identified specific genetic architectures that differentiate the perception of tinnitus from hearing loss³⁴. Additionally, evidence from neuroimaging studies points to distinct patterns of brain functioning associated with tinnitus^{36,37}. Moreover Edvall et al. recently demonstrated that altered auditory brainstem responses distinguish constant tinnitus from occasional tinnitus⁹. These findings underscore the importance of incorporating a broader range of environmental and biological factors in future studies.

3. Imputing missing data using feature medians can introduce biases, especially in a dataset as large and heterogeneous as the UKB. This simplistic approach to missing data warrants more critical reflection. Consider either applying more sophisticated imputation methods (e.g., multiple imputation) or providing a sensitivity analysis to demonstrate how different methods of handling missing data might affect the results. This is especially important given the complex interplay between various socio-demographic and health-related factors in the tinnitus models.

Response: We thank the reviewer for this suggestion, which was also raised by reviewer 1 (see our response to Reviewer 1 point #3 above).

4. The identified risk factors, such as mood, neuroticism, and sleep, are consistent with the literature, but they only explain a small proportion of the variance in tinnitus severity (3–7%). This suggests that other important risk factors remain unidentified. The manuscript could benefit from a more critical discussion on this point. For instance, should future studies explore genetic or neurobiological factors in more depth? The brief mention of genetics in the discussion seems insufficient, given the emerging evidence on the genetic contribution to tinnitus.

Response: The predictive power of the study seems weak when measuring the total variance explained between levels of tinnitus severity, as small changes between mild forms of tinnitus severity were not well accounted by the model. The model, however, performed very well for the prediction of individuals who initially reported no severe tinnitus and transitioned to severe forms of tinnitus. We argue that identifying the individuals at risk of evolving towards these most severe forms of tinnitus is what is the most clinically useful. In this specific case, our simplified POST model showed very good performance in a group of held out individuals (Cohen’s $d = 1.3$, ROC = 0.78). This was further validated in the TRI datasets (Cohen’s $d = 2.5$, ROC = 0.94), suggesting that our POST questionnaires could be performed in populations outside the UKBiobank. It is finally unclear if including biological variables would further improve an already excellent model (at least for severe forms of tinnitus), or if we would face a ceiling effect in performance.

Our new discussion was significantly expanded on biological factors contributing beyond the psychosocial factors identified in this study page 6. Thus, we agree with the reviewer that biological

factors like genetic and neurobiological factor should be explored in more depth to better understand tinnitus presence but also the severity. The model however showed good performance for identifying the individuals that will transition towards strong worsening of their condition (AUC > 0.8 in holdout participants), suggesting that they could benefit from early targeted intervention.

“Third, adding important biological factors in our analysis, such as genetic and cerebral contributions^{34,35}, should be considered in the future, especially for predicting presence of tinnitus as recent studies highlight the significant role of genetics. For instance, Clifford et al. identified specific genetic architectures that differentiate the perception of tinnitus from hearing loss³⁴. Additionally, evidence from neuroimaging studies points to distinct patterns of brain functioning associated with tinnitus^{36,37}. Moreover Edvall et al. recently demonstrated that altered auditory brainstem responses distinguish constant tinnitus from occasional tinnitus⁹. These findings underscore the importance of incorporating a broader range of environmental and biological factors in future studies.

5. The manuscript acknowledges the lack of ethnic diversity in the UKB sample (91% white). However, this issue needs a more critical exploration, as it limits the generalizability of the findings to non-white populations. In particular, the study should address whether tinnitus risk factors might differ across ethnic groups, and how future studies could overcome this limitation. Similarly, the differences in tinnitus prevalence and severity between men and women are noted, but further exploration into the gendered nature of these risk factors is missing. Could there be biological or psychosocial mechanisms that explain why men report more tinnitus but women report higher distress?

Response: Sex difference showed higher prevalence of tinnitus in men but greater severity in women. In the UKBiobank, we observe that men reported more hearing difficulties than women (t-test: $p < 10^{-360}$), and that women reported more depressive symptoms (Frequently tired: t-test: $p = 10^{-183}$), neuroticism (mood swings, t-test: $p = 10^{-187}$) and insomnia (t-test: stats= 29, $p = 10^{-183}$) than men. The observed sex differences in tinnitus prevalence could thus be due to the fact that men have higher risk factors for tinnitus presence (more difficulties in hearing health), and women have higher risk factors for tinnitus severity (more mental health and sleep difficulties). We next tested the validity of our findings separately in men and women (supplementary Figure S5) but observed that the multivariate risk factors were equivalent between sexes (see response to reviewer 2 point #3 above).

We further tested our models on different ethnicities (supplementary Figure S6; shown below) and found that the prediction generalized across different groups, suggesting that despite the small sample size, similar risk factors explained tinnitus presence or tinnitus severity across sexes and ethnicities.

6. While the POST questionnaire is a useful clinical tool, the manuscript falls short of explaining its real-world applicability. How will this tool be integrated into clinical practice? Will it be used in primary care settings, or is it more suited for specialized clinics? Moreover, there is little discussion of the potential limitations of using a questionnaire-based tool, especially in populations with low health literacy or in clinical settings where tinnitus may be underreported.

Response: This is an excellent point. We have now added a larger discussion on the possible use of the questionnaire in the discussion, while mentioning the need to clinical validation

“To improve clinical utility, we developed a 6-item questionnaire to predict tinnitus severity over time. Our findings show that higher scores on the questionnaire are associated with a larger odd-ratio of developing severe tinnitus. Notably, the questionnaire performs well in identifying out-of-sample participants at risk of severe tinnitus in the future, even if they report low distress at the time of completing it. These results were validated using an independent clinical dataset (TRI database, 462 participants), ensuring the generalizability of the findings. Overall, this tool represents an easy-to-use prognostic resource for identifying patients unlikely to habituate to tinnitus. This questionnaire has the potential to be a key tool in improving tinnitus clinical management. Clinical resources for tinnitus management are limited, including the number of ENTs specialized in tinnitus and therapists trained in cognitive behavioral therapy (CBT). Clinicians in primary care could use our questionnaire to help in their decisions to refer patients to tinnitus specialists. Additionally, specialists could use the tool to focus their clinical and therapeutic efforts on patients at higher risk of developing severe tinnitus, avoiding unnecessary or excessive interventions for those likely to habituate. This concern is particularly relevant in common clinical scenarios, such as idiopathic sensorineural hearing loss, the complex aftermath of otological surgery, or newly onset tinnitus. Further clinical studies are necessary to validate the efficacy of this tool across various clinical populations.”

Minor comments:

1. Some key references on tinnitus risk factors, particularly from recent meta-analyses and systematic reviews, are missing. For instance, the discussion on hearing health as the main predictor of tinnitus presence could be enriched by including more comprehensive literature reviews. Additionally, consider citing more recent work on the role of stress and mental health in tinnitus severity.

Response: Thank you for this remark. We included those references in the discussion:

- Biswas R, Genitsaridi E, Trpchevska N, et al. Low Evidence for Tinnitus Risk Factors: A Systematic Review and Meta-analysis. *JARO* 2023; **24**: 81–94.
- Cresswell M, Casanova F, Beaumont RN, et al. Understanding Factors That Cause Tinnitus: A Mendelian Randomization Study in the UK Biobank. *Ear & Hearing* 2022; **43**: 70–80.
- Jafari Z, Baguley D, Kolb BE, Mohajerani MH. A Systematic Review and Meta-Analysis of Extended High-Frequency Hearing Thresholds in Tinnitus With a Normal Audiogram. *Ear & Hearing* 2022; **43**: 1643–52.
- Kleinstäuber M, Weise C. Psychosocial Variables That Predict Chronic and Disabling Tinnitus: A Systematic Review. In: Searchfield GD, Zhang J, eds. *The Behavioral Neuroscience of Tinnitus*. Cham: Springer International Publishing, 2020: 361–80.
- Trevis KJ, Mclachlan NM, Wilson SJ. A systematic review and meta-analysis of psychological functioning in chronic tinnitus. *Clinical Psychology Review* 2018; **60**: 62–86.

- *Elarbed A, Fackrell K, Baguley DM, Hoare DJ. Tinnitus and stress in adults: a scoping review. International Journal of Audiology 2021; 60: 171–82.*

2. Figures, particularly those related to model performance (e.g., ROC-AUC curves), could benefit from clearer annotations. Readers unfamiliar with machine learning or statistical modeling may find the figures challenging to interpret. For instance, the difference between “tinnitus presence” and “tinnitus severity” in the figures could be more explicitly marked. It would also be helpful to label key features driving the models directly on the graphs.

Response: Thank you for this remark. We added subtitle in the figure, color coded differently tinnitus presence and severity, and improved the figures legend to help the comprehension for people who are less familiar with this type of analysis.

3. Although the manuscript mentions some limitations, such as the dataset's lack of diversity and the inability to predict tinnitus presence evolution, these issues are not explored with enough depth. The manuscript could benefit from a more thorough exploration of why these limitations exist and how future research could address them.

Response: We have included a larger discussion on the limitations of our study page 6

“Our study has several limitations. First, the UK Biobank lacks ethnic diversity, with 91% of our sample being of White descent. This limitation may introduce bias into our models, potentially leading to a mischaracterization of non-White participants³². Even if we confirmed that our models performed well for individuals of Asian and Black ethnicities (Supplementary Figure S6), replication in datasets with greater ethnic diversity is necessary to validate our findings. Second, the assessment of tinnitus presence lacks granularity in the UK Biobank, as it does not include a pure constant tinnitus category. Additionally, the absence of hyperacusis evaluation - a hypersensitivity to noise commonly co-occurring with tinnitus - limits our ability to account for this important comorbidity³³. Third, adding important biological factors in our analysis, such as genetic and cerebral contributions^{34,35}, should be considered in the future, especially for predicting presence of tinnitus as recent studies highlight the significant role of genetics. For instance, Clifford et al. identified specific genetic architectures that differentiate the perception of tinnitus from hearing loss³⁴. Additionally, evidence from neuroimaging studies points to distinct patterns of brain functioning associated with tinnitus^{36,37}. Moreover Edvall et al. recently demonstrated that altered auditory brainstem responses distinguish constant tinnitus from occasional tinnitus⁹. These findings underscore the importance of incorporating a broader range of environmental and biological factors in future studies.”

We sincerely thank the three reviewers and the editors for their constructive feedback and their positive evaluation of our work.

Reviewer #2 (Remarks to the Author):

Authors have adequately addressed my questions and concerns, congratulations for performing the additional analysis, which now provide a very much complete view on tinnitus!

I encourage the authors to consider applying to funds such as the ATA and BTA for implementing a risk calculator using a combination of the data from the UKBB and the new questionnaire to be used by either the public or clinics (see <https://qrisk.org/> - they provide such online tools for cardiovascular risks). This will provide a fantastic advancement for public awareness, risk prevention, and improved care.

Response: We sincerely thank the reviewer for its enthusiasm and constructive recommendations

Reviewer #4 (Remarks to the Author):

Overall, the authors have addressed the comments with detailed descriptions. They have provided proper justifications for their methodological choices, For instance, their use of median imputation for missing data and the selection of features for the models are now justified properly. Furthermore, they have conducted additional analyses, including sex-stratified evaluations and validations across diverse ethnic groups, which enhanced the generalizability and robustness of their findings.

The authors also mentioned the limitations of their study, including the dataset's lack of diversity, the absence of hyperacusis evaluation, and the limited granularity in tinnitus presence assessment. They provided thoughtful discussions on these points and proposed actionable steps for future research, such as integrating genetic, neurobiological, and other biological factors to improve the predictive power of their models. They have provided an expanded discussion on the applicability of the POST questionnaire in clinical settings, which is crucial for implementing and translating research findings.

Moreover, the authors have revised and enhanced the figures, annotations, and supplemental materials to provide a better understanding of the results. They have sufficiently addressed all the reviewer 3 comments.

Response: We sincerely thank the reviewer for its positive comments

Reviewer #5 (Remarks to the Author):

Having carefully read the manuscript, reviewer 1's comments and the authors' response, I believe that the authors have satisfactorily addressed the reviewer's comments.

I think this is a very good paper. The main limitation is the lack of discussion on how to use the tinnitus severity prediction tool in clinical practice, given that its predictive ability is far from

optimal and it is not clear what kind of intervention should be applied to subjects at higher risk. This should be more explicitly acknowledged in the limitations of the paper.

Response: We sincerely thank the reviewer for its positive comments. We added this sentence in the limitation section to answer its comment

“Finally, we developed a tool to identify individuals at risk of developing severe tinnitus; however, further research is needed to determine the most effective treatments to prevent this invalidating condition in those patients.”